# Differential cell autonomous responses determine the outcome of coxsackievirus infections in murine pancreatic α and β cells

Laura Marroqui[1†], Miguel Lopes[1†], Reinaldo S dos Santos[1†], Fabio A Grieco[1], Merja Roivainen[2], Sarah J Richardson[3], Noel G Morgan[3], Anne Op de beeck[1*], Decio L Eizirik[1*]

[1]ULB Center for Diabetes Research, Medical Faculty, Université Libre de Bruxelles, Brussels, Belgium; [2]National Institute for Health and Welfare, Helsinki, Finland; [3]Institute of Biomedical and Clinical Sciences, University of Exeter Medical School, Exeter, United Kingdom

**Abstract** Type 1 diabetes (T1D) is an autoimmune disease caused by loss of pancreatic β cells via apoptosis while neighboring α cells are preserved. Viral infections by coxsackieviruses (CVB) may contribute to trigger autoimmunity in T1D. Cellular permissiveness to viral infection is modulated by innate antiviral responses, which vary among different cell types. We presently describe that global gene expression is similar in cytokine-treated and virus-infected human islet cells, with up-regulation of gene networks involved in cell autonomous immune responses. Comparison between the responses of rat pancreatic α and β cells to infection by CVB5 and 4 indicate that α cells trigger a more efficient antiviral response than β cells, including higher basal and induced expression of STAT1-regulated genes, and are thus better able to clear viral infections than β cells. These differences may explain why pancreatic β cells, but not α cells, are targeted by an autoimmune response during T1D.

*For correspondence: anne.op. de.beeck@ulb.ac.be (AO); deizirik@ulb.ac.be (DLE)

†These authors contributed equally to this work

Competing interests: The authors declare that no competing interests exist.

## Introduction

Type 1 diabetes (T1D) is an autoimmune disease in which pancreatic β cells are targeted by a protracted attack by the immune system. This leads to death of most of the β cells while neighboring α cells survive. The disease is characterized by pancreatic islet inflammation (insulitis) and progressive β cell loss by apoptosis (*Ziegler and Nepom, 2010*; *Morgan et al., 2014*). The triggering of T1D probably depends on environmental factors that interact with predisposing genes to induce autoimmunity against the β cells (*Eizirik et al., 2009*; *Santin and Eizirik, 2013*; *Morgan and Richardson, 2014*; *Morgan et al., 2014*). Among the potential environmental factors, epidemiological, clinical, and pathological studies in humans support the implication of viral infections, particularly enteroviruses (e.g., coxsackievirus; CVB), as triggers for the development of T1D (*Helfand et al., 1995*; *Dotta et al., 2007*; *Yeung et al., 2011*; *Morgan and Richardson, 2014*; *Richardson et al., 2014*). CVB-specific antibodies and enteroviral RNA are more frequently observed in serum samples from T1D patients than in healthy individuals (*Helfand et al., 1995*; *Lonnrot et al., 2000*), and staining of human pancreatic islets revealed that the enteroviral capsid protein VP1 is present at a higher frequency in insulin-containing islets from patients with recent-onset T1D when compared to healthy controls (*Dotta et al., 2007*; *Richardson et al., 2013*). A meta-analysis of 33 prevalence studies involving 1931 T1D cases and 2517 controls confirmed a clinically significant association between enteroviral infections and islet autoimmunity and T1D in humans (*Yeung et al., 2011*). Additionally, it has been shown that the presence of enterovirus

**eLife digest** Type 1 diabetes is caused by a person's immune system attacking the cells in their pancreas that produce insulin. This eventually kills off so many of these cells—known as beta cells—that the pancreas is unable to make enough insulin. As a result, individuals with type 1 diabetes must inject insulin to help their bodies process sugars. One of the mysteries of type 1 diabetes is why the beta cells in the pancreas are killed by the immune system while neighboring alpha cells, which produce the hormone glucagon, are spared.

Scientists suspect a combination of genetic and environmental factors contributes to type 1 diabetes. Certain viruses, including one called Coxsackievirus, appear to trigger type 1 diabetes in susceptible individuals. Other factors may also make these individuals more likely to develop the disease. For example, they may 'express' genes that are thought to increase the risk of type 1 diabetes, many of which control how the immune system responds to viral infections. These genes may make susceptible individuals experience excessive inflammation, because inflammation is what ultimately kills off the beta cells.

Now, Marroqui, Lopes, dos Santos et al. provide evidence that suggests why the alpha cells are spared the immune onslaught in type 1 diabetes. In initial experiments, clusters of cells—known as islets—from the human pancreas were either exposed to small proteins that cause inflammation or infected with the Coxsakievirus. Both events caused a similar increase in the expression of particular immune response genes in the islets. This indicates that these islet cells are able to react to the virus and trigger a first line of defense, which will be further boosted when the immune system is subsequently called into action.

Islets contain both alpha and beta cells, and so further experiments on alpha and beta cells from rats investigated whether the two cell types respond differently when infected by the Coxsakievirus. The results revealed that alpha cells boost the expression of the genes needed to clear the virus to a greater extent than the beta cells, and so respond more efficiently to the virus. Therefore, an infection is more likely to establish itself in the beta cells and consequently trigger inflammation and the immune system's attack on the cells.

These observations explain one of the puzzling questions in the diabetes field and reinforce the possibility that a long-standing viral infection in beta cells—which seem to have a limited capacity to clear viral infections—may be one of the mechanisms leading to progressive beta cell destruction in type 1 diabetes. This knowledge will help in the search for ways to protect beta cells against both viral infections and the consequent immune assault.

RNA (*Oikarinen et al., 2011*) or antibodies anti-CVB1 (*Laitinen et al., 2014*) in blood can predict the development of T1D.

Insulitis is established and exacerbated in the context of a 'dialog' between pancreatic β cells and the immune system, regulated by the local production and release of chemokines and cytokines. These proteins attract and stimulate cells of the immune system, such as macrophages and cytotoxic T lymphocytes (*Eizirik et al., 2009*; *Willcox et al., 2009*; *Arif et al., 2014*). The immune cells cause selective β cell destruction both directly and via the production of pro-inflammatory cytokines, such as interleukin-1β (IL-1β), type II interferon (IFNγ), and tumor necrosis factor α. These cytokines are released closely to the target cells and modulate the expression of complex gene networks in β cells, leading to the release of chemokines and eventually to the activation of the intrinsic pathway of cell death in β cells (*Eizirik et al., 2009*; *Gurzov and Eizirik, 2011*). A key unanswered question in the field is why highly differentiated and specialized pancreatic β cells express these immune-related pathways.

Host defense in vertebrates is usually viewed as the task of specialized immune cells. This perception underestimates the capacity of many non-immune cells to trigger 'self-defense' or 'cell autonomous immune responses' against infection. These mechanisms are pre-existing in many cell types and can be up-regulated upon virus infection (*Yan and Chen, 2012*; *Randow et al., 2013*). This is an ancient form of cellular protection, present both in bacteria and metazoans. In vertebrates, for instance, cellular self-defense synergizes with innate and adaptive immunity to fight infections (*Randow et al., 2013*). The cell susceptibility/resistance of highly differentiated and poorly proliferating cells, such as neurons and pancreatic β cells, to microbial infection is a major determinant of clinical outcome.

It has been shown that cerebellum granule cell neurons and cortical neurons (CNs) have unique self-defense programs that confer differential resistance to infection by West Nile virus. Specifically, higher basal expression and faster up-regulation of IFN-induced genes improves the survival of granule cell neurons infected by West Nile virus (*Cho et al., 2013*). These responses rely on detection of microbial signatures by pattern recognition receptors (PRRs) (*Randow et al., 2013*). Several candidate genes for T1D expressed in human islets, such as the RIG-like receptor *MDA5* (*Colli et al., 2010*) and the regulators of type I IFNs *PTPN2* and *USP18* (*Moore et al., 2009*; *Colli et al., 2010*; *Santin et al., 2012*), modulate viral detection, antiviral activity, and innate immunity. The candidate genes described above (*Moore et al., 2009*; *Colli et al., 2010*; *Santin et al., 2012*) and CVB5 infection (*Colli et al., 2011*) regulate β cell apoptosis via activation of the BH3-only protein Bim. These observations support the concept that genetically modulated self-defense responses in β cells might play an important role in determining the outbreak of insulitis and the progression to T1D in face of viral infection or other stimuli (*Santin and Eizirik, 2013*).

Against this background, we have presently evaluated the global gene expression of cytokine-treated and virus-infected human islet cells, observing that these two treatments lead to similar up-regulation of a large number of genes, gene networks, and transcription factors involved in cell autonomous immune responses. This conclusion generated two additional questions, namely whether this self-defense response is islet cell specific and, if yes, whether these putative cellular differences may explain the preferential β cell targeting by the autoimmune assault. To answer these questions, we next compared the responses of FACS-purified rat pancreatic α and β cells to infection by potentially diabetogenic CVB5 and CVB4. The results obtained indicate that α cells trigger a more effective antiviral response than β cells, including higher basal and induced expression of STAT1-regulated genes, and are thus able to better clear viral infections as compared to β cells.

## Results

### Exposure of human islets to pro-inflammatory cytokines or infection by CVB5 induces expression of a similar network of cell autonomous-related immunity genes

We used previous microarray and RNA sequencing (RNAseq) analysis made by our group to compare the global gene expression of CVB5-infected human islets, evaluated by microarray analysis 48 hr after viral infection (HV) (*Ylipaasto et al., 2005*), against the gene expression of human islets exposed to the pro-inflammatory cytokines IL-1β + IFNγ, evaluated either by microarray analysis at 24, 36, or 48 hr (HC1) (*Lopes et al., 2014*) or by RNAseq at 48 hr (HC2) (*Eizirik et al., 2012*), focusing the analysis on over-expressed genes (*Figure 1*). Comparison of human islets exposed to cytokines and analyzed by either microarray or RNAseq showed a strong similarity in the top 20% ranked genes (50% common genes; *Figure 1*). Comparison between CBV5-infected human islets against cytokine-treated human islets indicated a large number of common genes, in particular among the top 20% genes (30–50% common genes). Interestingly, the area under the curve (AUC) for a comparison between different batches of human islets exposed to cytokines and analyzed either by microarray or RNAseq analysis was 0.209 (subtracted by a null area of 0.5), while the AUC for the comparisons virus vs cytokines (microarray vs microarray or microarray vs RNAseq) was, respectively, 0.154 and 0.127, that is, 74% and 61% of the cytokines vs cytokines comparison, indicating a close similarity between human islet cell responses to virus or cytokines. To exclude that these similarities were the result of non-specific cell stress responses, we compared the viral-induced gene expression (*Ylipaasto et al., 2005*) against genes modified by palmitate (HP) (*Cnop et al., 2014*), a metabolic stress unrelated to the immune response. There was limited similarity between virus- and palmitate-induced genes, with a curve close to random (*Figure 1*) and an AUC of 0.027, that is, <20% of the area observed when comparing virus- against cytokine-induced genes.

The superposition of up-regulated genes between virus-infected and cytokine-treated human islets confirmed the similarity between virus- and cytokine-induced genes, with >40% of the genes up-regulated in common between both islet treatments (*Figure 1—figure supplement 1*). By further comparing the RNAseq data obtained in human islets (*Eizirik et al., 2012*; *Lopes et al., 2014*) against known PRRs and other antiviral/antibacterial factors (*Randow et al., 2013*; *Carty et al., 2014*), we noticed that cytokine-exposed human islets highly express and up-regulate antiviral (e.g., TLR3, MDA5, RIG-1, APOBEC36, SAMHD1, TRIM22, CNP, Tetherin, Viperin etc) factors, while antibacterial factors

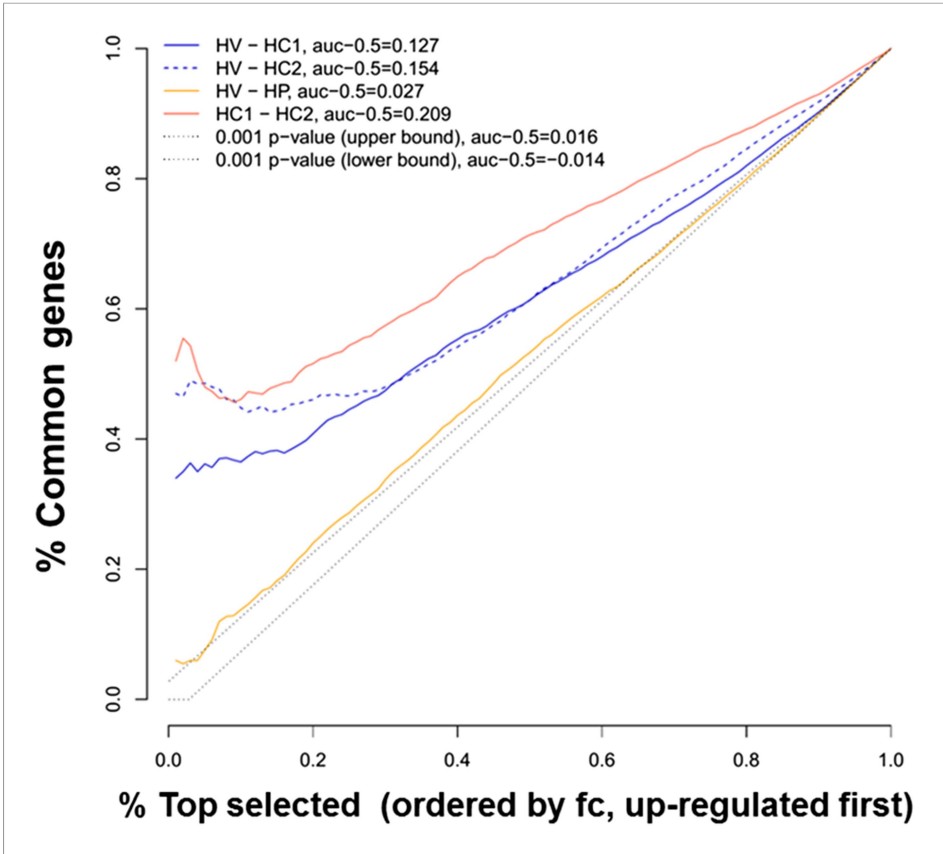

**Figure 1**. Ranking similarity between gene expression of human islets after cytokine exposure (HC1 and HC2) or after virus exposure (HV). The similarity between HC1 and HC2 and between HV and palmitate exposure (HP) is also presented. The area under the curve (subtracted by a null threshold of 0.5) is indicated, as well as similarity curves corresponding to a p-value of 0.001. The figure represents the ranking similarity ordered by up-regulation. (For a detailed explanation of the calculations done, see 'Materials and methods', 'gene expression ranking similarity').

The following figure supplements are available for figure 1:

**Figure supplement 1**. Venn diagram of the up-regulated genes in human islets exposed to cytokines (HC1 and HC2) or infected with CVB5 (HV).

**Figure supplement 2**. IPA analysis of the up-regulated genes in human islets exposed to cytokines HC1 or infected with CVB5 (HV).

**Figure supplement 3**. IPA analysis of the up-regulated genes in human islets exposed to cytokines HC2 or infected with CVB5 (HV).

(e.g., TLR4, NLRP1, CLEC6A, CLEC7A.) are lowly or not expressed, suggesting that these cells are under evolutionary pressure to counteract viral but not bacterial infections, probably because they are seldom confronted by bacteria. Pathway analysis of genes induced in common by cytokines and viral infection (*Figure 1—figure supplement 2* and *Figure 1—figure supplement 3*; for clarity only the top 30 correlations are shown) indicated presence of large groups of genes involved in interferon signaling (top correlation for both cytokine data sets compared to the virus data set), activation of IRFs (family of interferon regulatory transcription factors), PRRs, T1D signaling, role of PKR in IFN induction, Jak/Stat signaling (activated downstream of IFNs and STATs), and so on. These observations indicate that both a viral infection by CVB5 and exposure to pro-inflammatory cytokines trigger a cell autonomous immune response in human islet cells.

## Pancreatic α cells are resistant against CVB- but not against cytokine- or double-stranded RNA (dsRNA)-induced cell death

The experiments described above were performed with whole-human islets, composed to a large extent by β and α cells. Since it has been described that different brain cells present diverse innate immune response programs to viral infection (*Cho et al., 2013*), and since β but not α cells are killed in T1D, we next examined whether pancreatic β and α cells present different susceptibility to viral infections or to pro-inflammatory cytokines. It is presently not technically feasible to FACS-purify human β and α cells for long-term in vitro experiments due to the high-background fluorescence of human β cells caused by marked lipofuscin accumulation (*Cnop et al., 2010*) and the putative impact of antibody-mediated techniques on the long-term survival and function of the sorted cells. These follow-up experiments were thus performed in FACS-purified rat β and α cells (>90% pure cells and with >90% viability after 4 days in culture; there was also similar expression of the housekeeping gene GAPDH between α and β cells under different experimental conditions; *Figure 2*). Both β and α cells were killed to a same extent after a 24 hr exposure to different concentrations of IL-1β + IFNγ (*Figure 3A* and *Figure 3—figure supplement 1*), indicating that under the present experimental conditions FACS-purified rat α and β cells are similarly affected by cytokine-induced apoptosis. One of the mechanisms involved in cytokine-induced cell death is NO production (*Eizirik and Mandrup-Poulsen, 2001*), and β and α cells showed roughly similar iNOS expression and medium nitrite accumulation, with slightly higher nitrite production by β cells, but basal iNOS expression was higher in α than β cells (*Figure 3—figure supplement 2A,B*). Similarly, cytokines

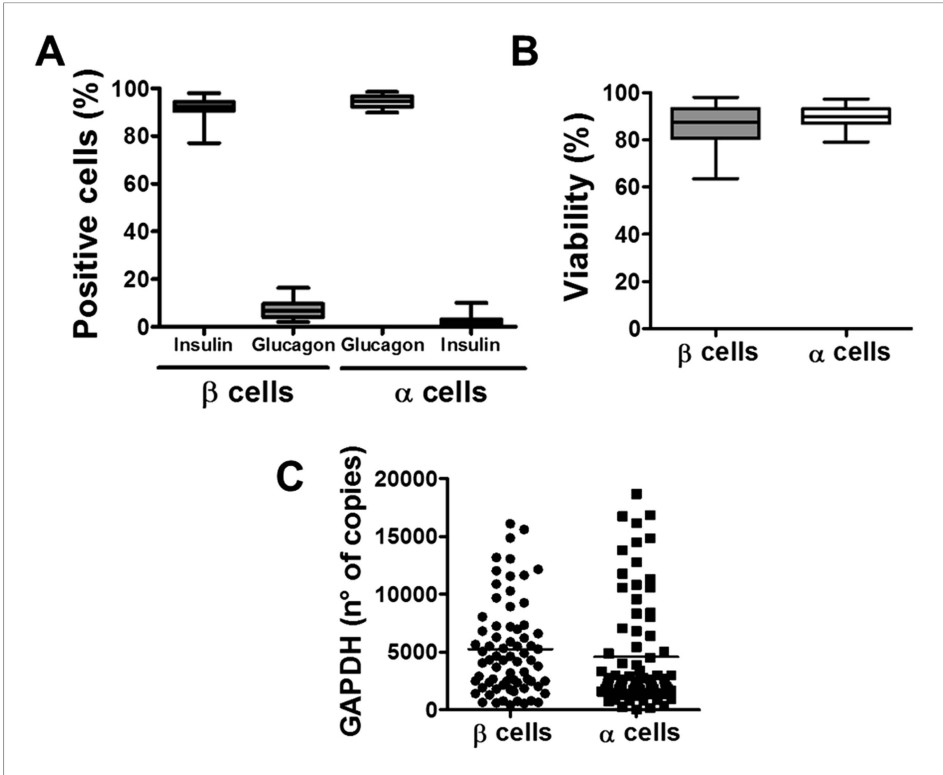

**Figure 2**. Purity, viability, and GAPDH mRNA expression of the β- and α-cell fractions after single-step FACS purification. (**A**) Immunostaining for insulin or glucagon of the rat islet cell preparations used in this study. Percentage of insulin- and glucagon-positive cells in the β and α cell preparations. (**B**) Cell viability was evaluated by staining the β and α cells with the nuclear dyes Hoechst 33342 and propidium iodide after 4 days in culture. Results are plotted as box plots, indicating lower quartile, median, and higher quartile, with whiskers representing the range of the remaining data points of 32 independent preparations. (**C**) GAPDH values were measured by RT-PCR and compared with a standard curve. Results are plotted as scatter dot plot of each single measurement (n = 70 for each cell type).

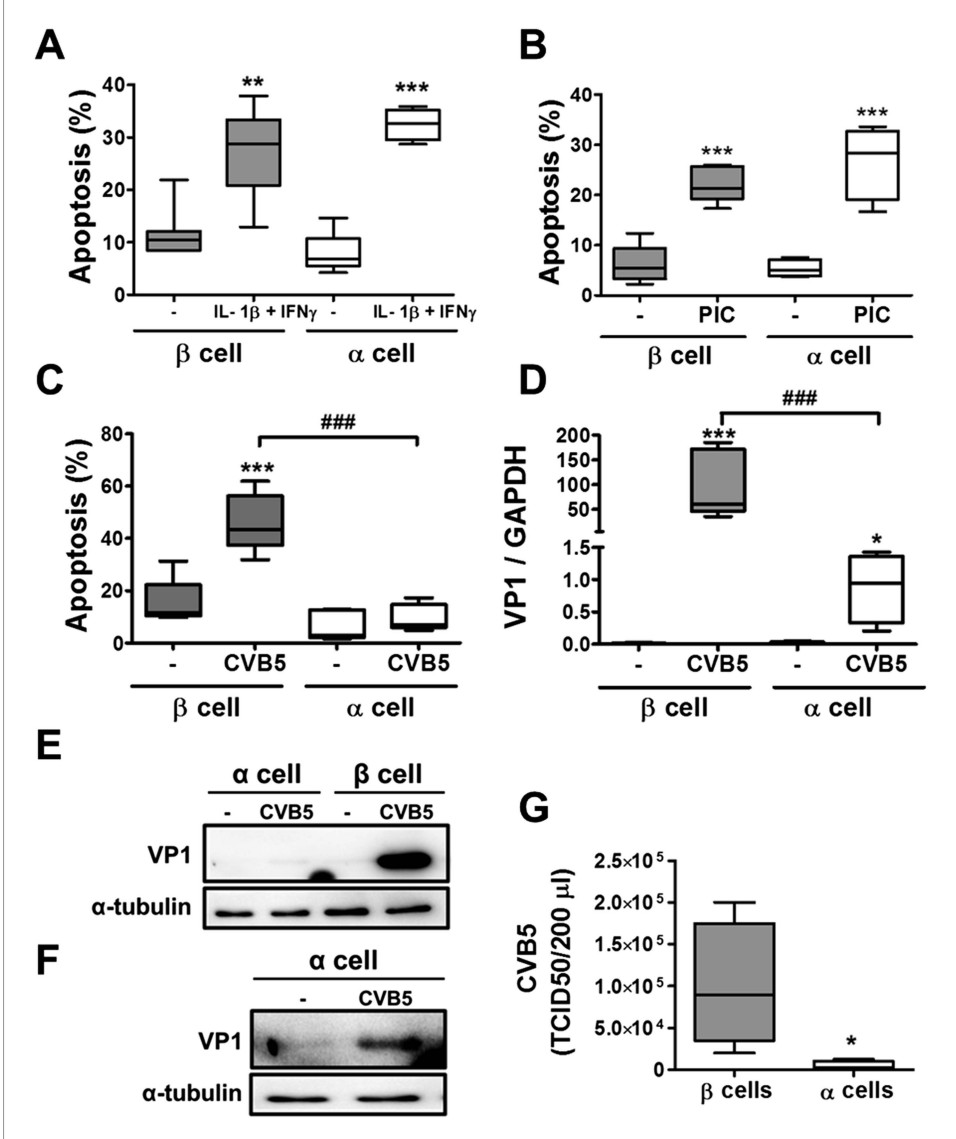

**Figure 3**. Pancreatic α cells are more resistant than β cells against CVB5- but not against cytokine- or PIC-induced cell death. FACS-purified rat β and α cells (>90% purity for both cell types) were treated with interleukin-1β (IL-1β) + type II interferon (IFNγ) (50 or 500 U/ml, respectively) (**A**) or PIC (1 μg/ml) for 24 hr or infected with CVB5 (multiplicity of infection—M.O.I. 5) for 36 hr (**C–G**). (**A–C**) Apoptosis was evaluated by staining with the nuclear dies Hoechst 33342 and PI. (**D**) VP1 mRNA expression was assayed by RT-PCR and normalized by the housekeeping gene GAPDH. (**E** and **F**) The figures show representative Western blots of VP1 protein expression after CVB5 infection and α-tubulin for loading control. (**F**) The Western blot (**E**) was overexposed to allow visualization of VP1 expression in α cells. (**G**) Titration of the supernatants from β and α cells infected with CVB5 for 36 hr. Results of 4–6 experiments are plotted as box plots, indicating lower quartile, median, and higher quartile, with whiskers representing the range of the remaining data points; *p < 0.05, **p < 0.01, and ***p < 0.001 treated vs untreated (**A** and **B**) or CVB5 vs mock infection (**C**, **D**, and **G**); ###p < 0.001 as indicated by bars; ANOVA followed by Student's t-test with Bonferroni correction.

The following figure supplements are available for figure 3:

**Figure supplement 1**. Dose-response of cytokine-induced apoptosis in pancreatic α and β cells.

**Figure supplement 2**. Pancreatic rat α and β cells have similar NO production, cytokine, and chemokine expression following exposure to cytokines.

*Figure 3. continued on next page*

*Figure 3. Continued*

**Figure supplement 3**. Pancreatic rat α and β cells have similar cytokine and chemokine expression following exposure to PIC.

**Figure supplement 4**. Prolonged time-course of CVB5-induced apoptosis in pancreatic α and β cells.

**Figure supplement 5**. UV-inactivated CVB5 does not induce cell death in pancreatic α and β cells.

**Figure supplement 6**. Cell counting after CVB5 infection of pancreatic α cells.

**Figure supplement 7**. Pancreatic α cells infected with CVB5 under different medium conditions.

induced expression of the chemokines CXCL10 and CCL2 in both cell types, but CCL2 expression was higher in α cells basally and following cytokine-exposure (*Figure 3—figure supplement 2C,D*). In line with the cytokine data, the synthetic dsRNA polyinosinic-polycitidilic acid (PIC) induced a similar percentage of cell death in β and α cells (*Figure 3B*), but a slightly higher expression of the downstream genes IFNα (*Figure 3—figure supplement 3B*) and CCL2 (*Figure 3—figure supplement 3F*) in α compared to β cells. Furthermore, basal expression of the transcription factor STAT1, iNOS and of the chemokines CXCL10 and CCL2 was also higher in α than in β cells (*Figure 3—figure supplement 3C,F*).

In contrast with the observations made with cytokines and dsRNA (see above), infection of these cells with CVB5 at a multiplicity of infection (M.O.I.) 5 induced death of nearly 50% of β cells after 36 hr, but it did not increase α cell death at this time point (*Figure 3C*) or after a more prolonged follow-up (up to 96 hr) post-infection (*Figure 3—figure supplement 4*). UV-inactivated virus did not kill β cells (*Figure 3—figures supplement 5*), indicating that cell death is the consequence of actual viral infection and proliferation. Additionally, there were no differences in the number of α cells attached to the well before and after the infection (*Figure 3—figure supplement 6*), supporting the assumption that the α cells remain alive after CVB5 infection. α cells resistance was independent of the glucose/fetal bovine serum (FBS) concentration used in the medium (*Figure 3—figure supplement 7*), and it was paralleled by a markedly lower expression of the CVB5 marker VP1 in α than in β cells, as evaluated by mRNA (*Figure 3D*) and protein (*Figure 3E,F*) expression, and by titration of the production of infective virus (*Figure 3G*). Dose-response experiments in α cells indicated that even at a M.O.I. of 100, CVB5 still induced markedly less cell death in α than in β cells infected at an M.O.I. 20-fold lower, that is, M.O.I. 5 (*Figure 4A*). This is not a phenomenon restricted to CVB5, since α cells were also much more resistant than β cells to CVB4-induced cell death (*Figure 4B*). This cannot be explained by different expression of receptors for the virus, since the CVB receptors coxsackievirus and adenovirus receptor (CAR) and decay accelerating factor for complement have similar or higher mRNA expression in rat α as compared to β cells (*Figure 4C,D*). The similar expression of CAR was confirmed at the protein level in α and β cells (*Figure 4—figures supplement 1*). Consistently, the same quantity of CVB5 remained associated with α and β cells after adsorption of the virus as measured by titration 2 hr after virus exposure (data not shown). In line with these findings, the percentage of infected α and β cells by a non-replicating adenoviral vector encoding GFP, which also enters the cells via CAR, was similar in α and β cells (*Figure 5A–C*). On the other hand, the intensity of GFP fluorescence was several-fold lower in α than in β cells (*Figure 5A,B,D*), indicating that the virus enters α cells but cannot properly translate its cargo protein. These observations may explain a common knowledge in the field that α cells are 'difficult to transduce' with adenoviral vectors.

## Pancreatic α cells express a vigorous cell-autonomous immune response against viral infection

We next compared the expression of known components of the cell-autonomous immune response in α and β cells under basal conditions (*Figures 6, 7*) and following infection with CVB5 (*Figure 8*). The cell preparations used were highly pure (>90%; *Figure 2A*), as confirmed by the 21-fold higher expression of glucagon (GCG) and the α cell transcription factor ARX in α cells as compared to β cells, and the lower expression of the β cell markers insulin (decrease of 185-fold) and PDX-1 (decrease of 12-fold) in these cells as compared to β cells (*Figure 6A*). In comparison to β cells, α cells have higher

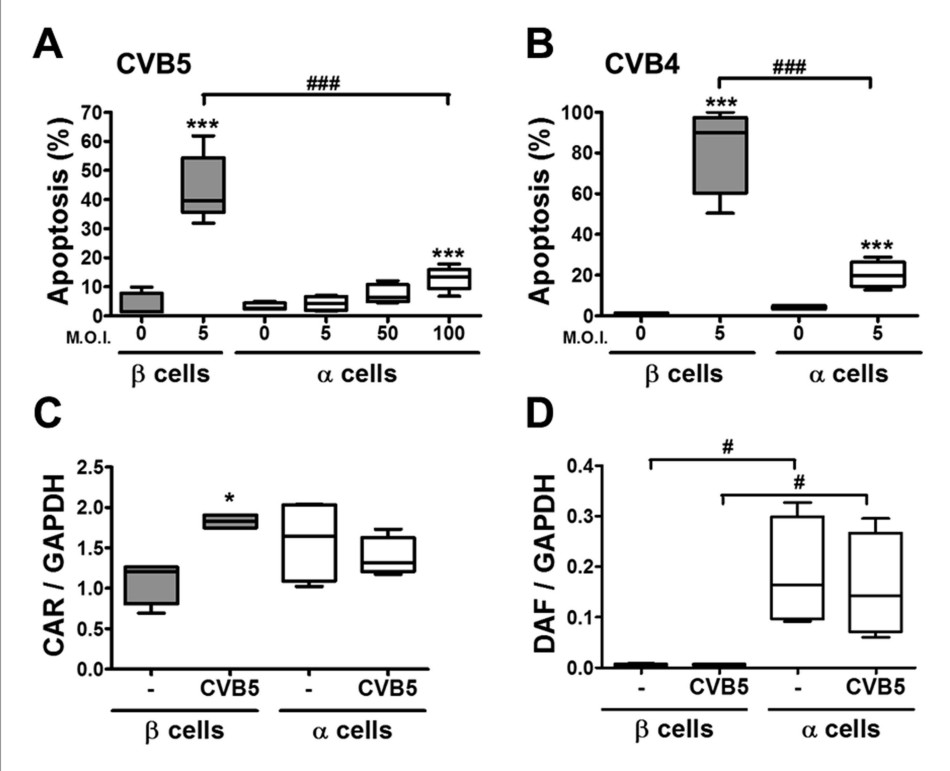

**Figure 4**. The higher susceptibility of β cells to virus-induced cell death, as compared to α cells, is not due to higher expression of virus receptors. FACS-purified rat α and β cells (>90% purity) were infected with CVB5 (M.O.I. 5, 50 or 100 for α cells; M.O.I. 5 for β cells) (**A**), CVB4 (M.O.I. 5) (**B**), or CVB5 (M.O.I. 5) (**C** and **D**) for 36 hr. (**A** and **B**) Apoptosis was evaluated by staining with the nuclear dies Hoechst 33342 and PI. Coxsackievirus and adenovirus receptor (CAR) (**C**) and DAF (**D**) mRNA expression were assayed by RT-PCR and normalized by the housekeeping gene GAPDH. Results are from 4–8 experiments, plotted as box plots indicating lower quartile, median, and higher quartile, with whiskers representing the range of the remaining data points; *$p < 0.05$ and ***$p < 0.001$ CVB5 or CVB4 vs mock infection; #$p < 0.05$ and ###$p < 0.001$ as indicated by bars; ANOVA followed by Student's t-test with Bonferroni correction. DAF, decay accelerating factor.

The following figure supplement is available for figure 4:

**Figure supplement 1**. CAR protein expression in pancreatic rat β and α cells.

basal expression of the chemokines CXCL10 and CCL2, of the cytokines IFNα and IL-1β (**Figure 6B**), of the transcription factor STAT1 (which mediates IFN signal transduction) and of several IFN-related downstream genes previously described to have a role in cell autonomous immune response in the brain (**Cho et al., 2013**), including Viperin, Tetherin, PKR, Mx1, USP18, Oas1, and so on (**Figure 6C**). Several of these genes have higher expression in granule cell neurons of the cerebellum, as compared to CNs, explaining why these cells are more resistant to infection by positive-stranded RNA viruses (**Cho et al., 2013**). There is remarkable similarity between gene expression in neurons and β cells (**Atouf et al., 1997**; **Villate et al., 2014**), and we next used available microarray and RNAseq data of, respectively, mouse granule neurons compared to CNs (**Cho et al., 2013**) and mouse α cells compared to β cells (**Benner et al., 2014**) to determine whether cells that have higher resistance to viral infection (i.e., granule neurons and α cells) have also increased basal expression of similar genes of the cell autonomous immune responses. The data shown in **Figure 7** and **Supplementary file 1** indicate a clear similarity between granule neurons and α cells for the genes present in both data sets; indeed, 64% of these genes were similarly increased in both cell types, with only 4% showing opposite direction of expression.

Following a 36 hr infection of α and β cells with CVB5 (**Figure 8**), α cells presented higher levels of the viral sensors MDA5 and PKR, of iNOS and CCL2, but lower induction of Viperin. A time-course analysis of α and β cells infected with CVB5 showed a progressive increase in the viral capsid protein

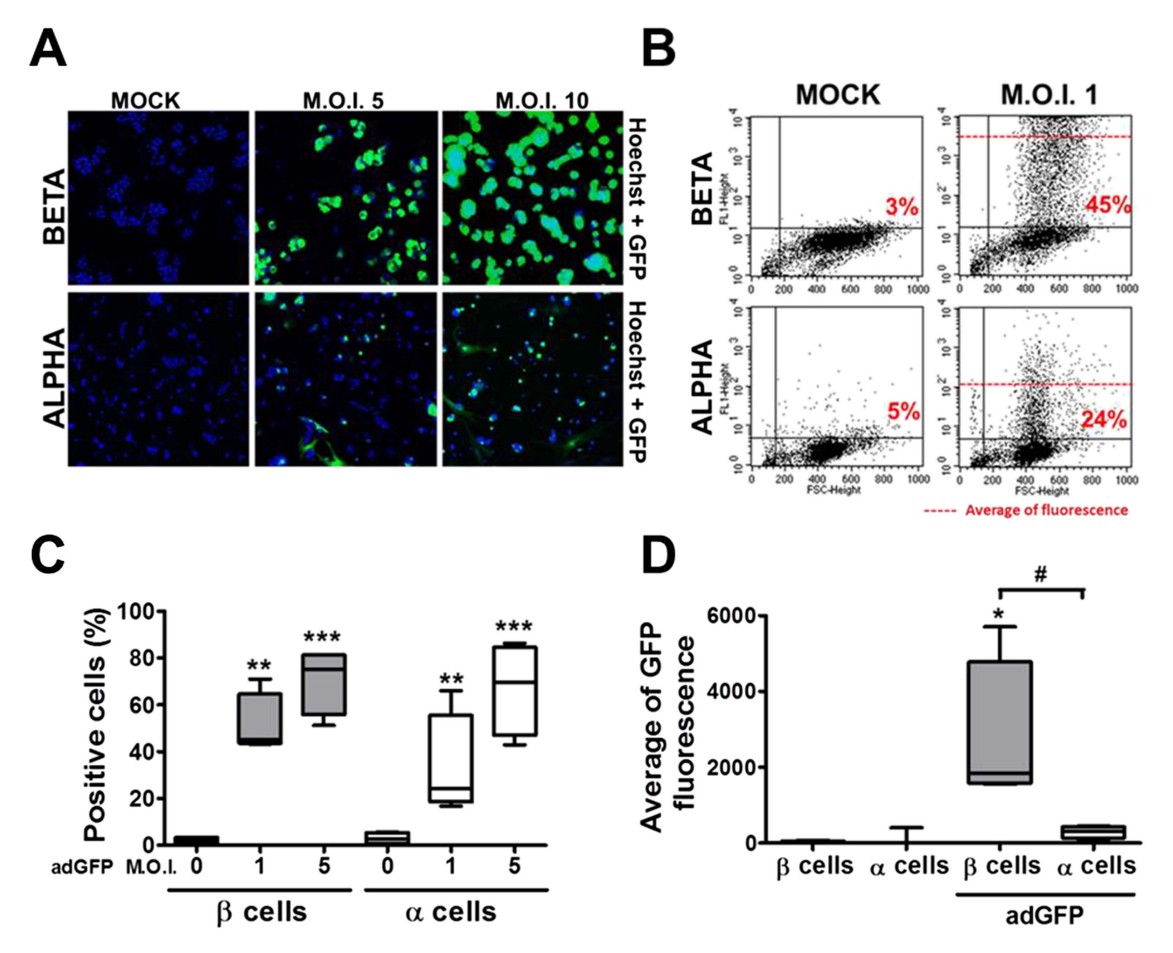

**Figure 5**. β and α cells are infected with similar efficiency by an adenoviral vector encoding GFP, but the translation of GFP protein is lower in α cells. (A–D) FACS-purified β and α cells (>90% purity for both cell types) were infected with adeno-GFP (M.O.I. 1, 5 or 10) for 48 hr. Presence of GFP protein was evaluated by fluorescence microscopy (A) and flow cytometry (B–D). (A) Pictures show nucleus (blue) and GFP fluorescence (green). β and α cells were infected with adeno-GFP M.O.I. 1 or 5 (B and D) for 48 hr and then sorted based on green fluorescence and forward-scattered light. (B) Representative 2-D plot of 4 independent experiments. (C) Quantification of GFP positive cells. (D) Average of green fluorescence intensity in cells infected at M.O.I. 5. Results of 4 experiments are plotted as box plots, indicating lower quartile, median, and higher quartile, with whiskers representing the range of the remaining data points; *p < 0.05, **p < 0.01 and ***p < 0.001 adeno-GFP vs mock infection; #p < 0.05 as indicated by bars; ANOVA followed by Student's t-test with Bonferroni correction.

VP1 in β cells (*Figure 9A*), while there was an early increase in VP1 expression in α cells (see inset in *Figure 9A*), with peak at 8 hr and subsequent decrease by 24 hr, suggesting that these cells are indeed infected by CVB5 but manage to eradicate the virus without dying. In line with this, α cells have both a higher basal expression and induction of two mRNAs encoding key antiviral proteins, namely STAT1 (*Figure 9B*) and MX1 (*Figure 9C*) as compared to β cells.

Additional experiments confirmed higher basal STAT1 mRNA (*Figure 10A*) and protein (*Figure 10B,C*) expression in α cells as compared to β cells. Knockdown of STAT1 in α cells by a previously validated (*Moore et al., 2011*) siRNA (*Figure 10D*) prevented MX1 up-regulation in response to CVB5 infection (*Figure 10E*) and enabled a more intense CVB5 infection, as indicated by increased VP1 expression (*Figure 10F*). These observations confirm that STAT1 plays a key role in α cell resistance to viral infection.

It has been previously shown that human β cells are more sensitive than α cells to CVB-induced infection and functional impairment both in vivo and in vitro (*Dotta et al., 2007*; *Richardson et al., 2009*; *Anagandula et al., 2014*), leading to the suggestion that CVB does not infect human α cells during the progression of T1D. Our present findings suggest an alternative hypothesis, namely that α

cells may become infected but are able to eradicate the virus more effectively than β cells (see *Figure 9*). If this hypothesis is correct, CVB infection should not be detectable in α cells of T1D patients when their islets are examined months/years after the putative initial infection. To test whether CVB may indeed infect human α cells, we first infected dispersed human islets from two donors (*Figure 11—figure supplement 1*) in vitro with CVB5 (M.O.I. 10) for 8 hr, the time point when maximum expression of VP1 was observed in a time-course experiment on α cells (*Figure 9*). We detected by immunofluorescence the presence of the enteroviral capsid protein VP1 in both α cells (glucagon-positive) and β cells (insulin-positive) (*Figure 11*). Thus, 52% (human islet sample 1) and 33% (human islet sample 2) insulin-positive cells were also positive for VP1, while 28% (human islet sample 1) or 27% (human islet sample 2) of cells were double positive for glucagon and VP1 (*Figure 11*). Similar results were observed with CVB4 (data not shown). To confirm these results in clinically relevant samples, we examined the pancreas of three children who died from myocarditis during the course of an acute and severe CVB infection. In these samples, we detected the presence of the enteroviral capsid protein, VP1, in β cells in all three cases and in α cells in two of three cases (*Figure 12* and *Figure 12—figure supplement 1*). In these latter patients, the number of infected α cells was clearly lower than the number of infected β cells.

## Discussion

A long-term puzzle in the pancreatic islet field has been why the highly specialized pancreatic β cells express a large number of immune-related genes upon exposure to pro-inflammatory cytokines (*Cardozo et al., 2001*; *Eizirik et al., 2009*, *2012*; *Ortis et al., 2010*; *Lopes et al., 2014*). From an evolutionary point of view, it would not make sense that these glucose-sensing and insulin-producing cells express these complex gene networks with the sole purpose to commit suicide during insulitis. The present observations, indicating a close similarity between cytokine- and virus-induced gene expression, suggest that these gene networks are actually part of a complex cell autonomous immune response, regulated at least in part by candidate genes for T1D (*Santin and Eizirik, 2013*), aiming to eradicate putative viral infections without excessive cell loss. Indeed, β cells, like neurons, have a very limited replication potential (*Cnop et al., 2010*), and an excessive loss of β cells would be disastrous for the host. In some genetically susceptible individuals, however, the putative initial viral infection might not be resolved leading to a persistent low-level infection, which in these individuals could trigger a specific autoimmune assault against β cells (*Dotta et al., 2007*; *Santin and Eizirik, 2013*; *Richardson et al., 2014*).

Pancreatic α and β cells are neighboring endocrine cells with a common embryonic origin (*Teitelman, 1993*). If an islet viral infection, which in theory should affect all islet cells, contributes to trigger T1D, an important question arises; namely, why are β cells killed while α cells survive? It has been previously suggested that α cells are more resistant to cytokine-induced apoptosis than β cells (*Mandrup-Poulsen et al., 1987*). The present observations, however, based on highly purified and viable rat α and β cells, cultured under similar conditions, indicate that both cell types are killed roughly to the same extent by the pro-inflammatory cytokines IL-1β + IFNγ. α and β cells are also equally susceptible to dsRNA-induced apoptosis, but α cells are several-fold more resistant to CVB-induced infection and consequent cell death than β cells. This suggests that the main difference between these two cell types is not their actual ability to detect signals of the viral infection (e.g., dsRNA) but how they respond to the virus once it is actively translating proteins inside the cells. In line with this hypothesis, both α and β cells were infected with an adenoviral vector encoding GFP, but α cells efficiently blocked (by 90%) the translation of virally-encoded GFP. A limitation of the present study is that the experiments described above were performed in FACS-purifed rat α and β cells, which limits extrapolation to the human disease.

Analysis of the expression of cell autonomous immunity-related genes in α and β cells (present data) and comparison between global expression of these genes in virus-resistant mouse granule cell neurons (*Cho et al., 2013*) and α cells indicate that a large number of IFN- and STAT1-dependent genes are expressed at higher level in α cells as compared to β cells. This suggests that this IFN-STAT1 gene network contributes to the resistance of α cells to viral infection. In support of this hypothesis, KD of STAT1 prevents CVB-induced MX1 expression in α cells and enables a more active infection with CVB5, as evaluated by higher expression of the enteroviral capsid protein VP1. Of note, unphosphorylated STAT1 supports the induction of antiviral genes in other cell types even without IFN-dependent stimulation (*Yang and Stark, 2008*; *Cheon and Stark, 2009*) and increased basal expression of STAT1 amplifies cell-intrinsic immune responses (*Hu et al., 2008*; *Amit et al., 2009*).

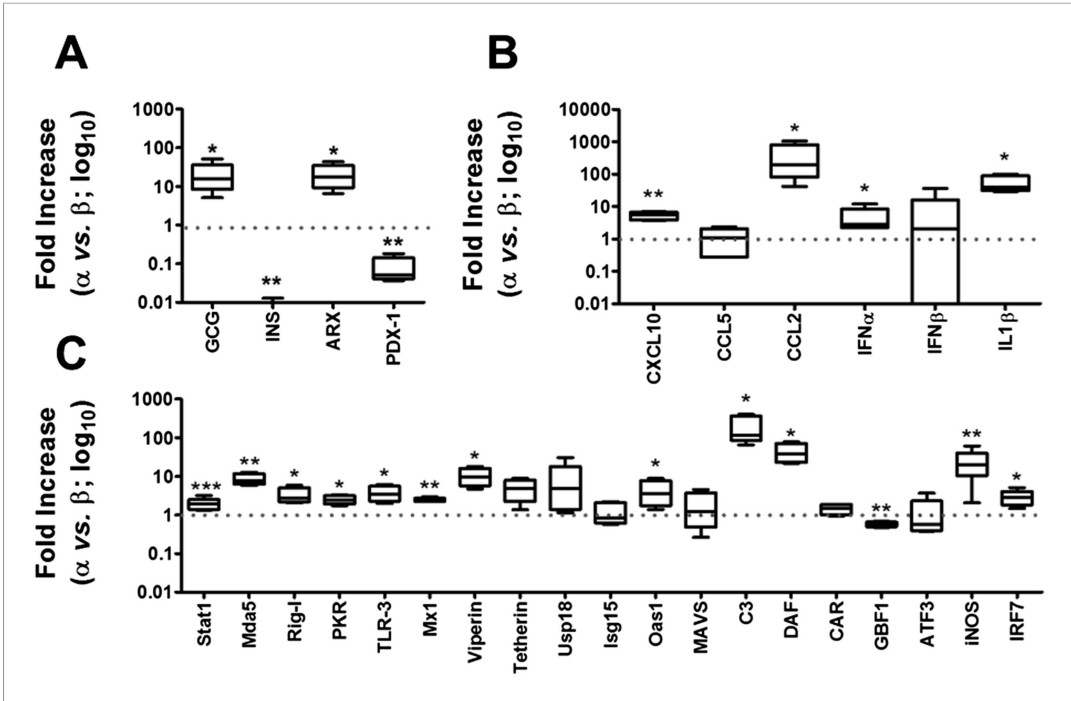

**Figure 6**. Basal expression of cell-autonomous immune response genes is higher in α cells than in β cells. mRNA expression of genes related to identity of β and α cells (**A**) or cell-autonomous immune response (**B** and **C**) was assayed by RT-PCR and normalized by the housekeeping gene GAPDH. Graphs represent relative expression of mRNAs in α cells vs β cells (dotted line indicates 1, i.e., no change). Results from 4–9 experiments are plotted as box plots, indicating lower quartile, median, and higher quartile, with whiskers representing the range of the remaining data points; *p < 0.05, **p < 0.01, and ***p < 0.001 α cells vs β cells; Student's t-test with Bonferroni correction.

Previous histological studies in pancreas from T1D individuals have shown that CVB is detected in β but not α cells (*Ylipaasto et al., 2004*; *Richardson et al., 2009, 2013*; *Anagandula et al., 2014*; *Morgan et al., 2014*), leading to the suggestion that CVB does not infect human α cells. Our present findings, obtained in highly purified α cells in vitro and by the analysis of dispersed human islets and pancreas from children with an acute and severe CVB infection, suggest an alternative hypothesis, namely that α cells indeed get infected but they rapidly eradicate the virus, probably due to their enhanced cell autonomous immunity responses.

Our proposal that a strong cell immune response protects α cells against viral infection and subsequent death in T1D seems at odds with the association of polymorphic variants of MDA5 that reduce helicase activity with a lower risk to develop T1D (*Nejentsev et al., 2009*; *Winkler et al., 2011*; *Lincez et al., 2015*). The long-lasting detection of enteroviral protein expression and IFN-induced markers in β cells in pre-diabetic or diabetic donors (*Dotta et al., 2007*; *Richardson et al., 2013, 2014*); however, most probably reflects a chronic non-cytolytic infection. In this context, the continuous stimulation of vigorous antiviral responses, in an attempt to control the infection, may lead to protracted presentation of β cell autoantigens, local release of chemokines and cytokines and eventually autoimmunity (*Santin and Eizirik, 2013*). This process, however, would not take place in α cells, which, as presently shown, have in place adequate mechanisms to swiftly eliminate the virus in the early stages of infection.

As mentioned above, pancreatic α and β cells have a common embryonic origin and are localized in the same microorgan (*Teitelman, 1993*). They thus provide a unique model to understand how molecular regulation of self-defense against viral infection in β cells, as compared to α cells, may determine cell death, local inflammation, and eventual diabetes. The present observations suggest that pancreatic α and β cells have different cell autonomous signatures. This may explain their different ability to clear viral infections and potentially explain why putative chronically infected pancreatic β cells, but not α cells, are targeted by an autoimmune response and killed during T1D.

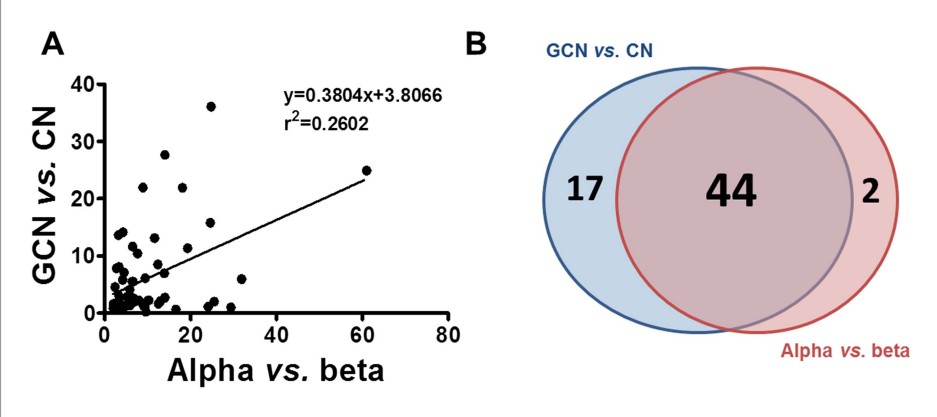

**Figure 7**. Similarity between up-regulated genes in granule neurons and pancreatic α cells. (**A**) Correlation between up-regulated genes in brain (cerebellum granule cell neurons (CGNs) vs cortical neurons [CNs]) and islet cells (alpha vs beta). (**B**) Venn diagram of the up-regulated genes in brain (CGN vs CN) and islets cells (alpha vs beta). The absolute values are shown in *Supplementary file 1*.

## Materials and methods

### Gene expression data sets

Gene expression after cytokine exposure was analyzed based on two data sets. The first (HC1) consists of 10 samples of human islets evaluated by RNAseq at 48 hr of IL-β + IFNγ exposure (*Eizirik et al., 2012*). The second (HC2) consists of 9 samples of human islets evaluated by microarray analysis at 24 hr, 36 hr, and 48 hr of IL-β + IFNγ exposure (*Lopes et al., 2014*). Gene expression after coxsackievirus exposure (HV) was evaluated in 3 samples of human islets evaluated by microarray analysis at 48 hr of CVB5 infection (*Ylipaasto et al., 2005*). The palmitate data set (HP) consists of 5 samples of human islets, evaluated by RNAseq at 48 hr of palmitate exposure (*Cnop et al., 2014*). All the samples are paired with their respective non-treated controls, that is, human islets obtained from the same donor and control condition (*Ylipaasto et al., 2005*; *Eizirik et al., 2012*; *Lopes et al., 2014*). A group of 9504 genes commonly identified in all data sets were considered for the analysis.

### Gene expression ranking similarity

In order to assess the similarity of gene expression in two different data sets, the following procedure was adopted. Gene expression fold change was calculated for each paired sample in each data set (both composed of the same genes). In each data set, genes were then ranked by mean fold change. A plot was drawn associating to each number of genes n (as a ratio to the total, x-axis) the number of common genes in the first n of the two rankings (divided by n). Lines corresponding to p-values thresholds (in this case 0.001) and the AUC are also presented (this statistic is known as the Sørensen–Dice index). AUC values and lines delimiting an area of null similarity (i.e., expected similarity of two random rankings) corresponding to a p-value of 0.001 (hypergeometric distribution) are also shown. Note that this analysis concerns similarity of up-regulation (genes are ranked from high- to low-fold change).

### FACS purification, culture, and treatment of rat β and α cells

Male Wistar rats (Charles River Laboratories, L'Arbresle Cedex, France) were housed and used according to the guidelines of the Belgian Regulations for Animal Care, with the approval by the local Ethical Committee (protocol number 465N; period of validity 07/2013-07/2017). Rat islets were isolated by collagenase digestion and hand picked. For β and α cells isolation, islets were dissociated into single cells by mechanical and enzymatic dispersion using trypsin (1 mg/ml) (Sigma, Bornem, Belgium) and DNase I (1 mg/ml) (Roche Applied Science, Indianapolis, USA) for 5 min at 31°C under agitation. Dissociated cells were re-suspended in HEPES-buffered Earle's medium containing 2.8 mM glucose and purified by FACS as described in (*Marroqui et al., 2015*). After sorting, purified β cells were cultured in Ham's F-10 medium containing 10 mM glucose, 2 mM GlutaMAX, 0.5% bovine serum

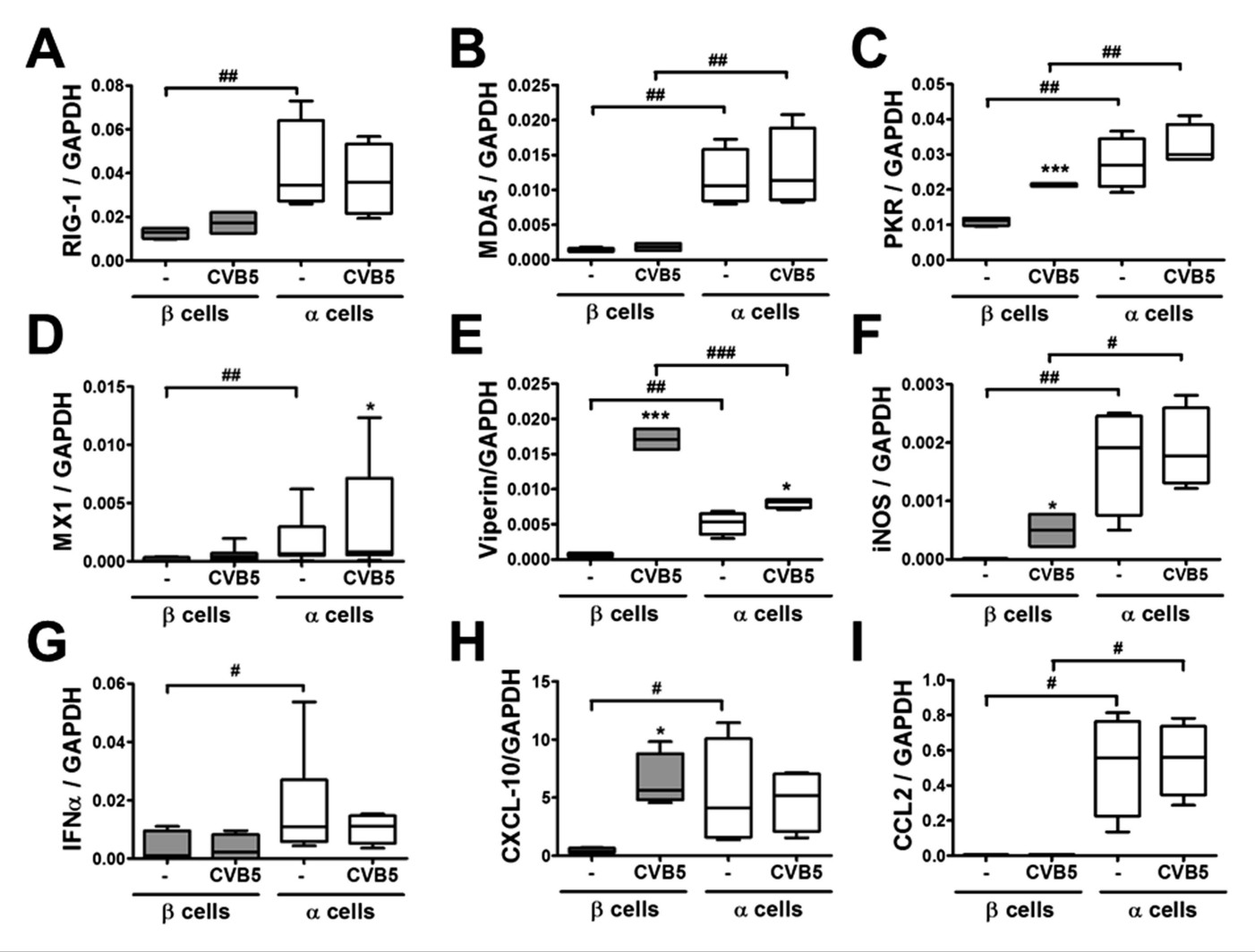

**Figure 8**. Differential expression of virus recognition and antiviral response genes in β and α cells exposed to CVB5. (**A–I**) FACS-purified β and α cells (>90% purity for both cell types) were infected with CVB5 (M.O.I. 5) for 36 hr. mRNA expression of genes related to virus recognition (**A–C**) and antiviral responses (**D–I**) was assayed by RT-PCR and normalized by the housekeeping gene GAPDH. Results for 4–8 experiments are plotted as box plots, indicating lower quartile, median, and higher quartile, with whiskers representing the range of the remaining data points; *p < 0.05 and ***p < 0.001 CVB5 vs mock infection; #p < 0.05, ##p < 0.01, and ###p < 0.001 as indicated by bars; ANOVA followed by Student's t-test with Bonferroni correction.

albumin (BSA), 50 µM isobutylmethylxanthine, 50 units/ml penicillin and 50 µg/ml streptomycin and 5% heat-inactivated fetal bovine serum (FBS, Gibco Life Technologies, Germany). α Cells were cultured in the same medium but with 6.1 mM glucose and 10% FBS. Purity of the β and α cell preparations was evaluated by immunofluorescence. α and β cells were immunostained with mouse monoclonal anti-insulin (Sigma, Bornem, Belgium) or mouse monoclonal anti-glucagon (Sigma, Bornem, Belgium) for 1 hr followed by rabbit anti-mouse secondary antibody conjugated with AlexaFluor 488 or AlexaFluor 567. The purity was calculated as a % of positive cells in each cell type (*Figure 2A*).

## Culture of human islets

Human islets were isolated from 2 non-diabetic organ donors (*Figure 11—figure supplement 1*) with approval from the local Ethical Committee in Pisa, Italy. Organ and tissue donation in Italy is regulated by the art. 23 of the national law n. 91, issued on 1 April 1999; in Tuscany the regional transplant organization (OTT, Organizzazione Toscana Trapianti) allows that organs not suitable for clinical transplantation are used for research purposes provided informed consent has been signed by the

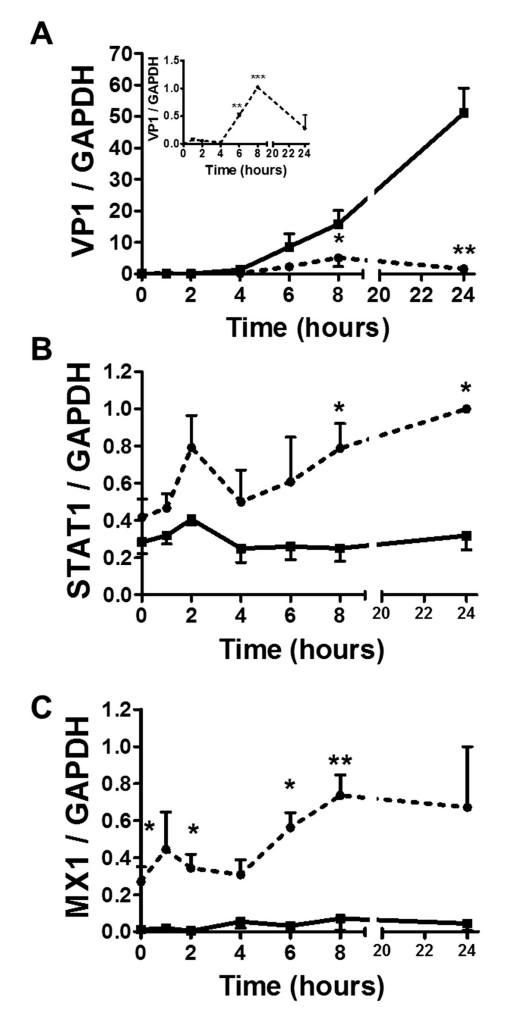

**Figure 9**. Time-course analysis of gene expression in β and α cells infected with CVB5. (**A**–**C**) FACS-purified β (squares and solid lines) and α cells (circles and dotted lines) were infected with CVB5 (M.O.I. 5) for 1, 2, 4, 6, 8, 24 hr. VP1 (**A**), STAT1 (**B**), and MX1 (**C**) mRNA expression were assayed by RT-PCR and normalized by the housekeeping gene GAPDH. Inset in 6A show details of VP1 expression in α cells. Results are mean values ± SEM of 3–4 independent experiments; *p < 0.05, **p < 0.01, and ***p < 0.001 α vs β; ANOVA followed by Student's t-test with Bonferroni correction. Inset; **p < 0.01 and ***p < 0.001 CVB5 vs mock infection; One-way ANOVA followed by Student's t-test with Bonferroni correction.

responsible relative. Prof Marchetti's group has access to donated pancreases for the preparation and study of isolated islets on the basis of approval by their local ethics committee, renewed in 2013. Isolation of human islets was done by collagenase digestion and density-gradient purification (*Marchetti et al., 2007*). Subsequently, isolated islets were cultured in M199 medium containing 5.5 mM glucose (*Marchetti et al., 2007*). Within 1–5 days of isolation, the human islets were shipped to Brussels. After arrival in Brussels and overnight recovery, the human islets were dispersed and cultured in Ham's F-10 medium containing 6.1 mM glucose, 2 mM GlutaMAX, 50 μM 3-isobutyl-1-methylxanthine, 1% charcoal-absorbed bovine serum albumin, 10% FBS, 50 mg/ml streptomycin, and 50 units/ml penicillin. The proportion of β cells and α cells in the preparations was determined by immunocytochemistry for insulin and glucagon, respectively (*Eizirik et al., 2012*).

## Cell treatments and nitric oxide measurement

Cells were treated with recombinant human IL-1β (R&D Systems, Abingdon, U.K.) and recombinant rat IFNγ (R&D Systems, Abingdon, U.K.) for 24 hr. The concentrations of cytokines (*Marroqui et al., 2014*) used are indicated in the figures.

The synthetic dsRNA analog PIC (Invitrogen, San Diego, CA, USA) was used at the final concentration of 1 μg/ml and its transfection into cells was performed under the same conditions as used for siRNA (see below) but using 0.15 ml of Lipofectamine 2000 (*Colli et al., 2011*).

Culture supernatants were collected for nitrite determination (nitrite is a stable product of nitric oxide [NO] oxidation) at OD540 nm using the Griess method (*Green et al., 1982*).

## Viral infection

The prototype strains of enterovirus (CVB5/Faulkner; CBV-4/J.V.B.) were obtained from American Type Culture Collection (Manassas, VA). This virus was passaged in Green Monkey Kidney cells. The identity of the enterovirus preparations used was confirmed using a plaque neutralization assay with type-specific antisera (*Roivainen et al., 2000*). We choose to analyze the effect of CVB5 and CVB4, both serotypes detected in T1D donors (*Yeung et al., 2011*), in order to allow comparisons with our previous studies based on infection of β cells with these viral serotypes (*Ylipaasto et al., 2004, 2005, 2012*). Importantly, both CVB4 and CVB5 have been detected in islets of T1D donors (*Yeung et al., 2011*) and CVB4 has been associated to diabetes onset (*Dotta et al., 2007*; *Gallagher et al., 2015*). We did not test the CVB1 serotype because it has been shown that CVB1 does not multiply in rat β cells (*Nair et al., 2010*).

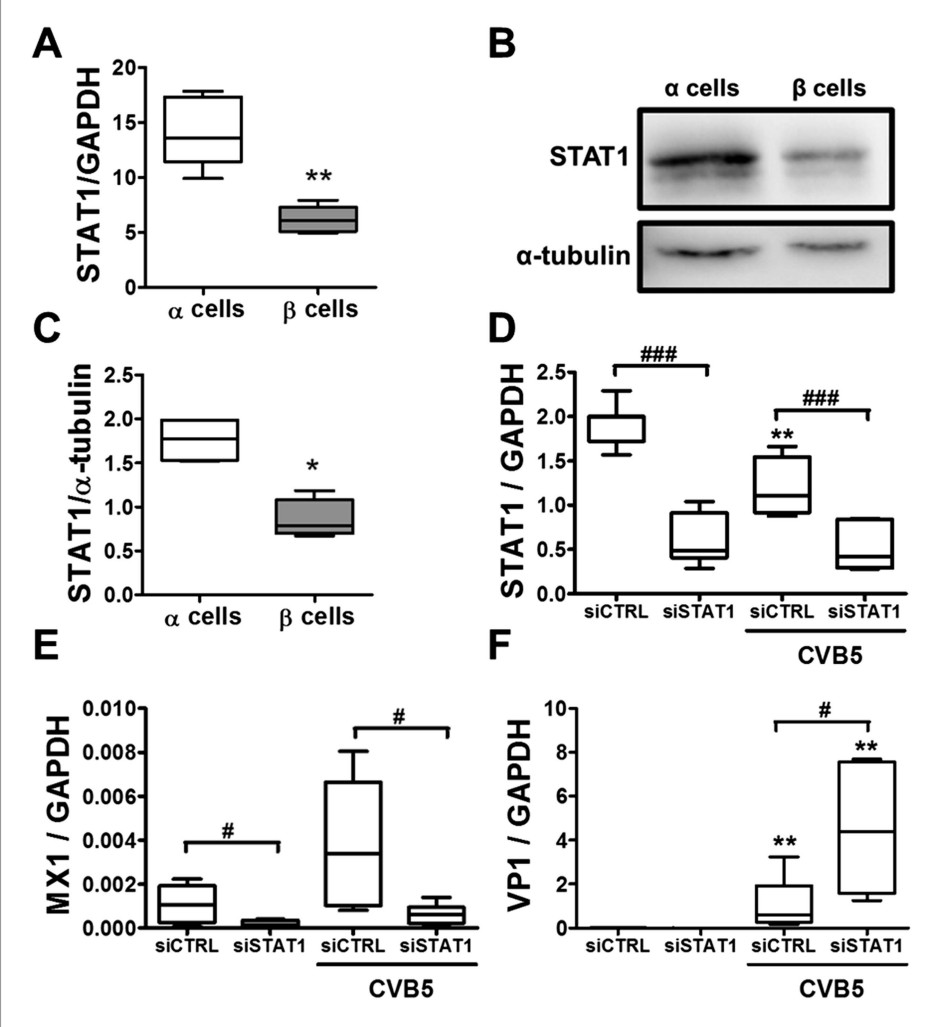

**Figure 10**. Knockdown of STAT1 decreases MX1 and increases VP1 expression in pancreatic rat α cells after CVB5 infection. (**A–C**) FACS-purified β and α cells (>90% purity for both cell types) were collected after 3 days in culture. Basal expression of STAT1 mRNA level (**A**) was assayed by RT-PCR and normalized by the housekeeping gene GAPDH. STAT1 protein expression was measured by Western blot (**B** and **C**). Densitometry quantification of 4 independent samples of each cell type is shown in (**C**). (**D–F**) FACS-purified α cells (>90% purity) were transfected with siCTRL or siSTAT1 (**D–F**). After 48 hr of recovery, cells were infected with CVB5 (M.O.I. 5) for 36 hr. STAT1 (**D**), MX1 (**E**), or VP1 (**F**) mRNA expression was assayed by RT-PCR and normalized by the housekeeping gene GAPDH. Results from 4–6 experiments are plotted as box plots, indicating lower quartile, median, and higher quartile, with whiskers representing the range of the remaining data points; *p < 0.05 and **p < 0.01 and CVB5 vs mock infection; #p < 0.05 and ###p < 0.001 as indicated by bars; ANOVA followed by Student's t-test with Bonferroni correction.

Viral stocks were prepared in GMK cells and titrated by plaque assay as previously described (*Roivainen et al., 2000*) or by limit dilution assay; viral titers obtained in plaque forming unit/ml and in 50% tissue culture infectious dose/ml were similar. β cells, α cells, and dispersed human islets were infected with virus diluted in β and α cells, or human islets medium in the absence of serum at the indicated M.O.I. After adsorption for 2 hr at 37°C, the inoculum virus was removed and cells were washed 3 times with medium. Serum-containing medium was added to the plates and the virus was allowed to replicate for indicated time periods. Inactivated virus was prepared by UV irradiation with a 1000J/m² dose and proper inactivation verified by titration on GMK cells.

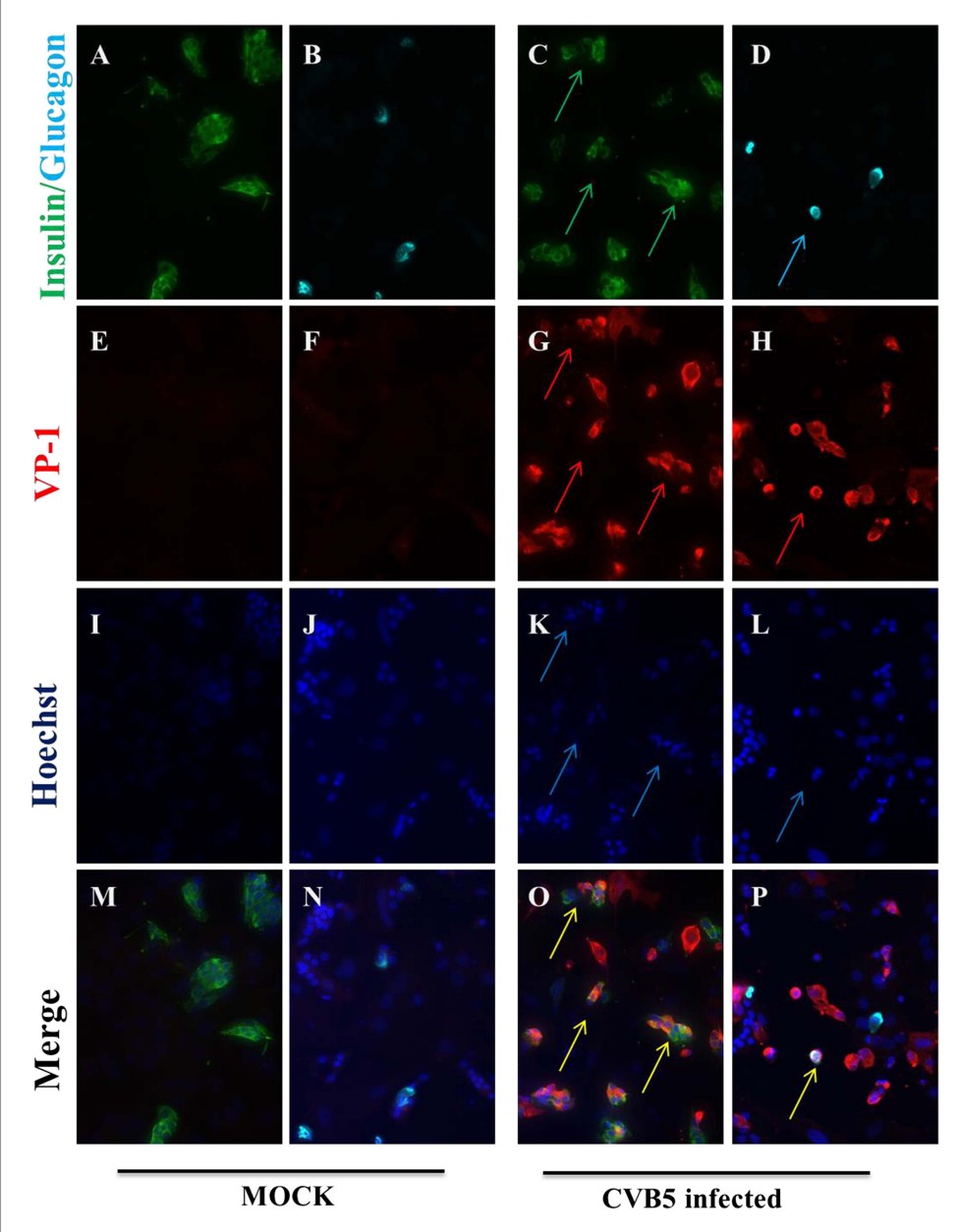

**Figure 11**. Infection of both α and β cells in dispersed human islets exposed to high titers of CVB5 for 8 hr. Dispersed human islets were mock infected or infected with CVB5 (M.O.I. 10) for 8 hr. After infection, cells were fixed and used for histological studies. Fluorescent microscopy analysis of insulin (**A** and **C**, in green), glucagon (**B** and **D**, in cyan), and VP-1(**E**–**H**, in red) shows the presence of double-positive cells for insulin and VP-1 (**O**, merged panels, in yellow) and glucagon and VP-1(**P**, merged panels, in yellow/white) after CVB5 infection. No VP-1 positive cells (**E** and **F**) were observed in mock-infected cells. Nuclear staining was performed with Hoechst (**I**–**L**, in blue). Double-positive cells for insulin and VP-1 and for glucagon and VP-1 are indicated by the arrows (**C**, **D**, **G**, **H**, **K**, **L**, **O**, and **P** panels).

The following figure supplement is available for figure 11:

**Figure supplement 1**. Characteristics of the 2 human donors used in the present study.

## Viral titration

Infected cells were frozen in their medium and thawed three times to release the virus. Total infectivity was assayed using end-point dilutions in microwell cultures of GMK cells. Cytopathic effects were read

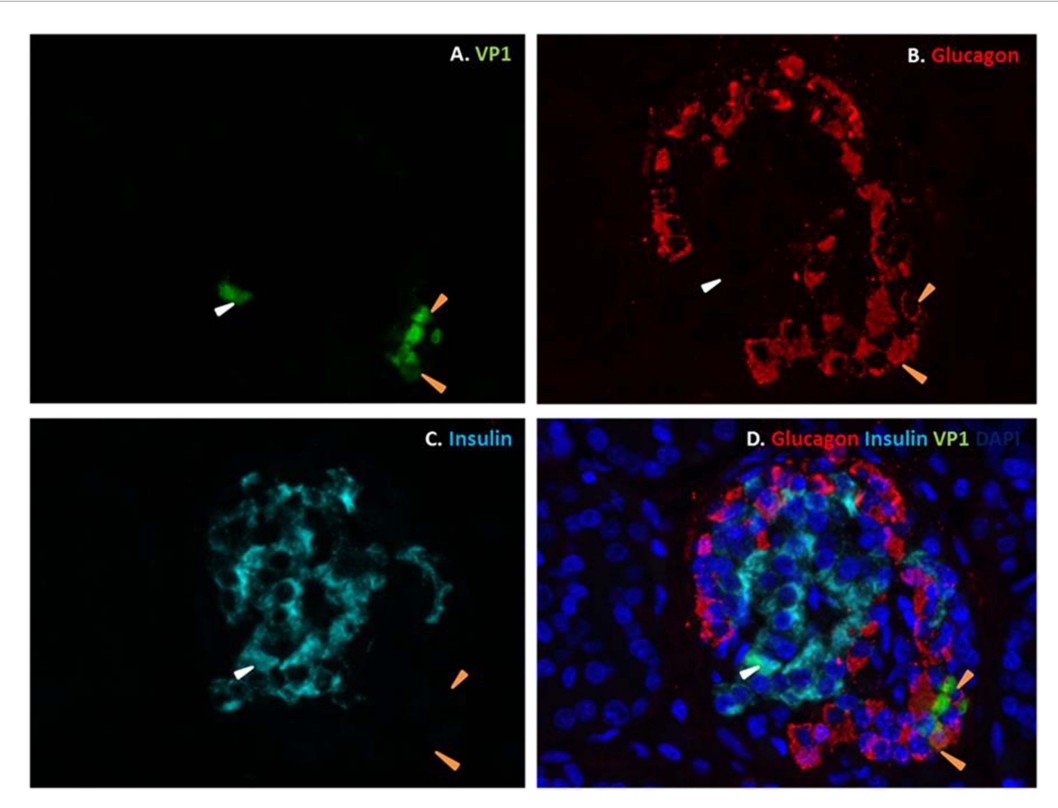

**Figure 12**. Fluorescence photomicrographs of an islet from a neonate with an acute coxsackievirus infection. Viral VP1 (green; **A**, **D**) co-localizes with glucagon (red; **B**, **D**) in certain cells (orange arrows) and with insulin (light blue; **C**, **D**) in another (white arrow). Nuclei were stained with DAPI (dark blue) in the merged image (**D**).

The following figure supplement is available for figure 12:

**Figure supplement 1**. List of human samples used.

on day 6 by microscopy, and 50% tissue culture infectious dose titers were calculated using the Kärber formula (*Lennette and Schemidt, 1969*).

## Adeno-GFP infection and flow cytometry

β and α cells were infected with an adenovirus encoding Green fluorescent protein (adeno-GFP; [*Heimberg et al., 2001*]) diluted in β or α cell medium in the absence of serum at M.O.I. 1, 5, or 10. After adsorption for 3 hr at 37°C, the inoculum was removed, serum-containing medium was added to the plates, and the cells were allowed to express GFP for 48 hr. Cells were detached with mild trypsin treatment and suspended in 2% paraformaldehyde-containing phosphate buffered saline (PBS). Cells were then analyzed on a flow cytometer (FacsCalibur, BD Biosciences, San Jose, CA). Analysis was performed using CellQuest Pro software version 6.0 (BD Biosciences, San Jose, CA). The cellular populations were selected based on size and cell granularity and analyzed for green fluorescence.

## RNA interference

α Cells were transfected with 30 nM of the previously validated siRNA for STAT1 (5′-CCCUAGAAGACUUACAAGAUGAAUA-3, Invitrogen, Carlsbad, CA, USA; [*Moore et al., 2011*]) or Allstars Negative Control siRNA (siCTRL, Qiagen, Venlo, the Netherlands; used as a negative control) using the Lipofectamine RNAiMAX lipid reagent (Invitrogen, Carlsbad, CA, USA). siCTRL does not affect α cell gene expression, function, or viability (data not shown). Cells were cultured for 48 hr after transfection and then infected with CVB5.

## Assessment of cell viability and cell counting

The percentage of viable, apoptotic, and necrotic cells was determined after incubation with the DNA-binding dyes propidium iodide (5 µg/ml; Sigma, Bornem, Belgium) and Hoechst 33342 (5 µg/ml; Sigma, Bornem, Belgium) (*Rasschaert et al., 2005*). A minimum of 600 cells was counted in each experimental condition. Viability was evaluated by two independent observers, one of them unaware of sample identity. The agreement between observers was >90%.

Cell counting of floating (i.e., in the supernatant) or attached cells was performed in Neubauer chambers, and each point was measured in triplicate by two observers, one of them unaware of sample identity.

## mRNA extraction and real-time PCR

Poly(A)+mRNA was isolated from primary rat β and α cells using the Dynabeads mRNA DIRECT kit (Invitrogen, Carlsbad, CA, USA), reverse transcribed, and amplified by real-time PCR using SYBR Green as described (*Rasschaert et al., 2005*). Quantitative real-time PCR was compared with a standard curve (*Overbergh et al., 1999*). Expression values were corrected for the reference gene glyceraldehyde-3-phosphate dehydrogenase (GAPDH), whose expression is not modified by the presently utilized experimental conditions (*Figure 2C* and data not shown). Primers are detailed in *Supplementary file 2*.

## Western blot analysis

Cells were washed with cold PBS and lysed in Laemmli buffer. Immunoblot analysis was performed with anti-STAT1 (1:1000; Santa Cruz, Dallas, USA), enterovirus-specific rabbit antiserum (1:1000; KTL-510), anti-CAR (1:100; Santa Cruz Biotechnology), anti-α-tubulin (1:5000; Sigma, Bornem, Belgium), and anti-β-actin (1:2000; Cell signaling). Membranes were exposed to secondary peroxidase-conjugated antibody for 1 hr at room temperature. Immunoreactive bands were revealed using the SuperSignal West Femto chemiluminescent substrate (Thermo Scientific, Rockford, USA) and detected using a Bio-Rad chemi DocTM XRS+ (Bio-Rad laboratories). The densitometry of the bands was evaluated using Image Laboratoty software (Bio-Rad laboratories).

## Human samples and immunofluorescence

Immunofluorescence was performed as described (*Gurzov et al., 2010*). Briefly, cells were plated on polylysine-coated cover slips, infected with CVB5 or CVB4 M.O.I. 10 for 8 hr, and fixed with 4% paraformaldehyde. After permeabilization with 0.3% Triton X-100, cells were incubated for 1 hr with an enterovirus-specific rabbit antiserum (1:1000; KLT-510), mouse monoclonal anti-insulin (1:1000; Sigma, Bornem, Belgium), or mouse monoclonal anti-glucagon (1:1000; Sigma, Bornem, Belgium). Alexa Fluor 568 goat anti-rabbit IgG or rabbit anti-mouse IgG and Alexa Fluor 488 goat anti-mouse IgG were, respectively, applied for 1 hr (1:1000). After nuclear staining with Hoechst, cover slips were mounted with fluorescent mounting medium (DAKO, Carpintera, USA), and immunofluorescence was visualized on a Zeiss microscope equipped with a camera (Zeiss-Vision, Munich, Germany). Images were acquired at 40× magnification and analyzed using AxiVision software.

For the histological study of clinical samples, three human pancreases removed at autopsy from neonatal patients (3–14 days) with fatal coxsackievirus infections were employed (*Figure 12—figure supplement 1*). The cases were selected randomly from within a previously described collection (*Foulis et al., 1990*; *Richardson et al., 2009*). Specimens had been fixed in buffered formalin or unbuffered formol saline, and they were all paraffin-embedded. The coxsackievirus infected human tissue samples were from a historical collection compiled in the 1980s, when fully informed consent was not required. They are held in the Glasgow Diabetes Biobank and were analyzed under authority of the UK Human Tissue Authority (licence number 12276). Ethical permission was granted by Greater Glasgow Clyde Research Ethics Committee (Ref: 10/S0704/25).

Serial sections (4 µm) were mounted on glass slides coated in (3-aminopropyl)-triethoxysilane (Sigma, Dorset, UK). Antigens were unmasked by heat-induced epitope retrieval in 10 mM citrate buffer pH 6.0. To examine the islet cell subtypes expressing VP1, triple immunofluorescence staining was performed. Sections were incubated with antisera with specificity for the enteroviral capsid protein, VP1 (Dako; 5D8/1) overnight at 1:500 (optimal for these specimens) and detected using the Life Technologies TSA kit#2 (AlexaFluor 488) as per the manufacturer's instructions. The sections were

washed and stained with rabbit anti-glucagon (Abcam; 1:4000) for 1 hr followed by goat anti-rabbit secondary antibody conjugated with AlexaFluor 555. Finally, sections were incubated with a guinea pig anti-insulin serum (DAKO; 1/600), which was detected with a goat anti-guinea pig secondary antibody conjugated with AlexaFluor 647. DAPI (1:1000, Invitrogen) was included in the final incubation to stain cell nuclei. Some slides were processed in the absence of primary antibody or with isotype control antisera to confirm the specificity of labeling. Sections were mounted in fluorescence mounting medium (Dako) under glass cover slips and examined on a Leica AF6000 Microscope. Multi-channel images were collected, processed, and analyzed using LAS AF software.

## Statistical analysis

Data are presented as mean values ± SEM or plotted as box plots, indicating lower quartile, median, and higher quartile, with whiskers representing the range of the remaining data points. Comparisons were performed by two-tailed paired Student's t-test or by analysis of variance (ANOVA) followed by Student's t-test with Bonferroni correction, as indicated. A p value <0.05 was considered as statistically significant.

## Acknowledgements

The authors thank I Millard, AM Musuaya, S Mertens, M Pangerl, and Marie-Louise Draps from the ULB Center for Diabetes Research, Université Libre de Bruxelles, for excellent technical support, Drs J-V Turantzine and O Villate, ULB Center for Diabetes Research, Université Libre de Bruxelles, for help in the RNA sequencing of human islets, and Drs P Marchetti and L Marselli, Pancreatic Islet Laboratory, University of Pisa, for providing the two human islets preparations used in the study. We are grateful to the Flow Cytometry Facility of the Erasmus Campus of the ULB and Christine Dubois for the cell sorting.

## Additional information

### Funding

| Funder | Grant reference | Author |
|---|---|---|
| Fonds De La Recherche Scientifique - FNRS | FNRS- F 5/4/5.MCF/KP. Project de secherche (PDR) T.0036.13. | Decio L Eizirik |
| European Commission (EC) | Projects Naimit and BetaBat, in the Framework Programme 7 of the European Community. | Decio L Eizirik |
| Fédération Wallonie-Bruxelles | the Communaute Française de Belgique-Actions de Recherche Concertees (ARC). | Decio L Eizirik |
| Fonds De La Recherche Scientifique - FNRS | FNRS post-doctoral fellowship. | Laura Marroqui |
| Governo Brasil | PDE/CSF Pós-Doutorado no Exterior | Reinaldo S dos Santos |
| Juvenile Diabetes Research Foundation International (JDRF) | JDRF Career Development Award. | Sarah J Richardson |
| European Commission (EC) | European Union's Seventh Framework Programme [FP7/2007-2013] under grant agreement 261441 PEVNET. | Sarah J Richardson, Noel G Morgan |

The funders had no role in study design, data collection and interpretation, or the decision to submit the work for publication.

## Author contributions
LM, AO, Contributed to the original idea and the design of the experiments, Researched data, Contributed to discussion, Revised and edited the manuscript; ML, RSS, FAG, MR, SJR, NGM, Researched data, Contributed to discussion, Revised and edited the manuscript; DLE, Proposed the original idea of the study and contributed to the design and interpretation of the experiments, Wrote the manuscript; He is the guarantor of this work and, as such, had full access to all the data in the study and takes responsibility for the integrity of the data and the accuracy of the data analysis

## Ethics
Human subjects: Human islets were isolated from 2 non-diabetic organ donors with approval from the local Ethical Committee in Pisa, Italy. Organ and tissue donation in Italy is regulated by the art. 23 of the national law n. 91, issued on 1 April 1999; in Tuscany the regional transplant organization (OTT, Organizzazione Toscana Trapianti) allows that organs not suitable for clinical transplantation are used for research purposes provided informed consent has been signed by the responsible relative. Prof. Marchetti's group has access to donated pancreases for the preparation and study of isolated islets on the basis of approval by their local ethics committee, renewed in 2013.

Animal experimentation: Male Wistar rats (Charles River Laboratories, L'Arbresle Cedex, France) were housed and used according to the guidelines of the Belgian Regulations for Animal Care, with the approval by the local Ethical Committee (protocol number 465N; period of validity 07/2013-07/2017).

# Additional files

## Supplementary files
• Supplementary file 1. Comparison between differentially expressed genes in granule cell neurons (compared to cortical neurons) and pancreatic α cells (compared to β cells).

• Supplementary file 2. List of primers used in the study.

## Major datasets
The following previously published datasets were used:

| Author(s) | Year | Dataset title | Dataset ID and/or URL | Database, license, and accessibility information |
|---|---|---|---|---|
| Eizirik DL, Cnop M, Bottu G | 2012 | The human pancreatic islet transcriptome: impact of pro-inflammatory cytokines | http://www.ncbi.nlm.nih.gov/geo/query/acc.cgi?acc=GSE35296 | Publicly available at the NCBI Gene Expression Omnibus (Accession no: GSE35296). |
| Kutlu B, Lopes M | 2013 | Human islets exposed to cytokines IL-1β and IFN-γ | http://www.ncbi.nlm.nih.gov/geo/query/acc.cgi?acc=GSE53454 | Publicly available at the NCBI Gene Expression Omnibus (Accession no: GSE53454). |
| Ylipaasto P, Kutlu B, Rasilainen S, Rasschaert J, Salmela K, Teerijoki H, Korsgren O, Lahesmaa R, Hovi T, Eizirik DL, Otonkoski T, Roivainen M | 2005 | Global profiling of coxsackievirus- and cytokine-induced gene expression in human pancreatic islets | http://dx.doi.org/10.1007/s00125-005-1839-7 | Publicly available at the Springer Link. |
| Cho H, Proll SC, Szretter KJ, Katze MG, Gale M Jr., Diamond MS | 2013 | Differential innate immune response programs in neuronal subtypes determine susceptibility to infection in the brain by positive-stranded RNA viruses | http://dx.doi.org/10.1038/nm.3108 | Available at the Nature Publishing Group (NPG). |

| Author(s) | Year | Dataset title | Dataset ID and/or URL | Database, license, and accessibility information |
|---|---|---|---|---|
| Benner C, van der Meulen T, Caceres E, Tigyi K, Donaldson CJ, Huising MO | 2014 | The transcriptional landscape of mouse beta cells compared to human beta cells reveals notable species differences in long non-coding RNA and protein-coding gene expression | http://dx.doi.org/10.1186/1471-2164-15-620 | Publicly available at the BioMed Central. |

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
