## [Decision Letter]

Thank you for sending your work entitled “Differential cell autonomous responses determine the outcome of coxsackievirus infections in pancreatic α and β cells” for consideration at *eLife*. Your article has been favorably evaluated by Tadatsugu Taniguchi (Senior editor), a Reviewing editor, and three external reviewers.

The Reviewing editor and the reviewers discussed their comments before we reached this decision, and the Reviewing editor has assembled the following comments to help you prepare a revised submission.

The reviewers identified two major concerns with the paper that were the subject of considerable discussion.

1) The first concerned the focus on rodent cells, rather than human cells which would potentially have more clinical relevance. The initial gene expression analysis is conducted in whole human islets (Figure 1 and some of the supplements), while the isolated β-cell and α-cell studies that form the crux of the paper are conducted in sorted rat islets (Figures 2, 3, 4, 5, 6, 7, 8, 9 and 10). The authors state that the reason for using rat cells in the majority of this work is technical: “It is presently not technically feasible to FACS-purify human β and α cells for long-term in vitro experiments due to the high background fluorescence of human β cells caused by marked lipofuscin accumulation”. The reviewers agreed that autofluorescence cannot be used, but it is not clear that β-cells and α-cells cannot be labeled with fluorescent proteins, similar to what has been done in many rodent studies. Recognising that this would be a considerable amount of work, the reviewers did not consider it essential that the work be repeated in human cells, but do require that any revision makes the limitations of working in rodent cells explicit.

2) The second concerned the relevance of the first part of the paper that makes a comparison between already published data sets on gene expression signatures in cytokine- or virus infected islets from humans. One of the reviewers felt that this was quite divorced from the rest of the paper and should be omitted (“nowhere else are cytokines used and the main part of the paper is neither supported nor refuted by these data”), but the others were more positive. On balance, we prefer that this section of the paper remains provided the authors can make clear how these data shed light on the substantive questions in the paper.

The reviewers also raised a number of comments that they considered less major but which are still substantive and require attention.

3) The authors state: “Both β and α cells were killed to the same extent after a 24 h exposure to IL-1β + IFNγ”. Is this what is commonly found? This reviewer was under the impression that multiple publications have shown that β-cells were more susceptible to such stresses. If not, perhaps it is related to the dose used here. Please include a more extensive dose-response study to confirm these findings.

4) There are a few overly generalized assumptions. For example, “we focused the analysis on over-expressed genes, since human islet genes down regulated after long-term exposure to stressful stimuli are often non-specific adaptive or apoptotic responses to cellular stress.” It is not clear whether this statement can be backed up with strong evidence. Please amend.

5) The time-course data are compelling, but in order to assess cell survival, time-courses would ideally be extended by several days. The life and death battle of islet cells can last 2-3 days. Ideally repeat the studies with longer exposure, otherwise explain why it was necessary to limit the duration of study.

6) The main conclusion of the paper is that α cells are only transiently infected with CBVs and can clear the virus infection, while β cells are infected without the ability to eradicate the virus. The stainings performed on pancreata from neonates who died from acute infections may in part support such a conclusion. The authors' conclusions would however be substantially supported if infections of whole pancreatic islets (which the authors should have at hand) would demonstrate that β cells die and α cells survive over time. Also, did the authors count the α cells before and after infection (e.g. 36h)? No change in cell numbers would support the conclusions drawn by the authors. Please provide the data requested.

7) On a similar note, α and β cells are, according to the Material and methods section, grown in different concentrations of FBS and glucose (α cells: 6.1 mM glucose and 10% FBS; β cells: β cells: 10 mM glucose and 5% FBS). Could the different culture conditions contribute to the differences in susceptibility to infection? Please comment on this, and if possible address.

8) The authors use CBV4 and CBV5 to infect cells. These viruses were grown in cell lines. An appropriate control in these experiments would be UV-inactivated virus, as cellular debris including cytokines etc. could have an effect on cell survival. Was such a control included? Please provide data, and if not, undertake studies to show that UV inactivated virus does not impact on cell survival.

9) By PCR it is demonstrated that both α and β cells express receptors for CBVs. More appropriate would have been to analyse surface expression by FACS, as mRNA expression. Please address this or explain why this is not possible.

Minor comments:

10) Provide a rationale for why you selected the CBV5 serotype for these experiments and not e.g. CBV1.

11) Virus concentrations are given as MOI, but the titers of replicating viruses in cells and media are given as TCID50 making these concentrations difficult to compare. Please, clarify.

---

## [Author Response]

*1) The first concerned the focus on rodent cells, rather than human cells which would potentially have more clinical relevance. The initial gene expression analysis is conducted in whole human islets (*Figure 1
*and some of the supplements), while the isolated β-cell and α-cell studies that form the crux of the paper are conducted in sorted rat islets (*Figures 2, 3, 4, 5, 6, 7, 8, 9 and 10*). The authors state that the reason for using rat cells in the majority of this work is technical:* “*It is presently not technically feasible to FACS-purify human β and α cells for long-term in vitro experiments due to the high background fluorescence of human β cells caused by marked lipofuscin accumulation*”*. The reviewers agreed that autofluorescence cannot be used, but it is not clear that β-cells and α-cells cannot be labeled with fluorescent proteins, similar to what has been done in many rodent studies. Recognising that this would be a considerable amount of work, the reviewers did not consider it essential that the work be repeated in human cells, but do require that any revision makes the limitations of working in rodent cells explicit*.

The concern raised by the editor and referees is well taken. The methods indicated by the editor and referees, i.e. labeling with fluorescent antibodies, recently used to sort α and β cells in pancreatic human islets (Dorrell et al., Stem Cell Res 183-194, 2008; Kirkpatrick et al., PLoS ONE e11053, 2010; Blodgett et al., Diabetes db150039, 2015) are indeed useful for studies involving RNA-seq or arrays of freshly isolated islet cells. They pose, however, a problem for the maintenance of the isolated cells in culture for several days after the purification, as required in the present study. Some of these methods require fixation of the cells, while others, where surface antibody are used without fixation, may affect cell function and trigger innate cellular responses secondary to antibody binding. Thus, and unfortunately, we presently cannot reproduce the current experiments based on viable and fully functional purified human β and α cells. In this context, the use of rodent FACS purified β and α cells provides a “second best” alternative. Indeed, Benner et al. assessed transcriptome-wide similarities and differences between rodent and human β cells, observing that 9905 genes are commonly expressed and 1540 differentially expressed between mouse and human β cells (Benner et al. BMC Genomics 620, 2014). In line with this study, a very recent paper compared the core proteomes of rat and human pancreatic β cells by label-free LC-MS/MS, quantifying more than 700 proteins belonging to key functional pathways, such as protein synthesis and intermediary metabolism (Martens GA, J Diabetes Res 549818, 2015). The authors describe that expression levels of these core functional pathways are highly conserved between rat and human β cells, despite some differences in proteins involved in glucose sensing pathways and redox control. On the other hand, we are well aware that working with rodent rather than human β cells present limitations, as these cells display structural and functional differences (Cabrera et al., Proc Natl Acad Sci USA 2334-2339, 2006; Henquin et al., Diabetes 3470-3477, 2006; Parnaud et al., Diabetologia 91-100, 2008).

We have now acknowledged the limitation of working with rat β and α cells in the Discussion.

*2) The second concerned the relevance of the first part of the paper that makes a comparison between already published data sets on gene expression signatures in cytokine- or virus infected islets from humans. One of the reviewers felt that this was quite divorced from the rest of the paper and should be omitted (*“*nowhere else are cytokines used and the main part of the paper is neither supported nor refuted by these data*”*), but the others were more positive. On balance, we prefer that this section of the paper remains provided the authors can make clear how these data shed light on the substantive questions in the paper*.

The comparison between gene expression signatures was a crucial step in the study, allowing us to realize that cytokine-treated β cells triggered a “cellular innate immune response”. This conclusion generated the second question of the study, namely whether this response is cell specific and, if yes, whether putative cell differences can explain why β cells, but not α cells, are eventually targeted and killed in T1D. We have now introduced a sentence in the Introduction, last paragraph, explaining how these data shed light on the key questions of the paper. Of note, I (DLE) have presented lectures based on these data at the EASD and Immunology of Diabetes Society meetings, and in both cases the audience understood well our strategy of starting with gene expression signatures of cytokine- and viral-infected islet cells and then evolving to the subsequent questions, as done in the present manuscript.

*The reviewers also raised a number of comments that they considered less major but which are still substantive and require attention*.

*3) The authors state:* “*Both β and α cells were killed to the same extent after a 24 h exposure to IL-1β + IFNγ*”*. Is this what is commonly found? This reviewer was under the impression that multiple publications have shown that β-cells were more susceptible to such stresses. If not, perhaps it is related to the dose used here. Please include a more extensive dose-response study to confirm these findings*.

The point is well taken. It has been indeed previously shown that α cells present higher resistance than β cells to proinflammatory cytokines, but these studies were mostly carried in mouse α and β cell lines, namely αTC1 and βTC1, respectively (Hamaguchi and Leiter, Diabetes 415-25, 1990; Iwahashi et al., Diabetologia 530-536, 1996; Barbagallo et al., BMC Genomics 62, 2013), and not in primary purified α and β cells, as in the present study.

Following the referees’ relevant suggestion, we have now performed a dose-response study using three concentrations of IL-1β + IFNγ, respectively 10 + 100 units/mL; 25 + 250 units/mL; and 50 + 500 units/mL. At these three different concentrations of cytokines we did not observe significant differences between α and β cell apoptosis (new Figure 3—figure supplement 1). These results reinforce the idea that, under the present experimental conditions, α and β cells are similarly susceptible to cytokines. We have now introduced a sentence in Results emphasizing that the present observations are related to specific experimental conditions, i.e. FACS purified rat α and β cells.

*4) There are a few overly generalized assumptions. For example,* “*we focused the analysis on over-expressed genes, since human islet genes down regulated after long-term exposure to stressful stimuli are often non-specific adaptive or apoptotic responses to cellular stress.*” *It is not clear whether this statement can be backed up with strong evidence. Please amend*.

This point is well taken. As suggested by the referees we have now removed the statement that “since human islet genes down regulated after long-term exposure to stressful stimuli are often non-specific adaptive or apoptotic responses to cellular stress”. This statement is actually based on our own unpublished bioinformatics analysis of RNAseq/microarray data of cytokine-, palmitate- and virus-infected β cells, but since these data is not yet publically available, we agree with the referees that it is better to remove the statement.

*5) The time-course data are compelling, but in order to assess cell survival, time-courses would ideally be extended by several days. The life and death battle of islet cells can last 2-3 days. Ideally repeat the studies with longer exposure, otherwise explain why it was necessary to limit the duration of study*.

To address this relevant question, we have now complemented our analysis by evaluating the impact of longer exposure of β and α cells to CVB5 (up to 96 h post infection). We observed a progressive increase in virus-induced β cell death, reaching 60-70% of cell death by 96 h. In contrast, most α cells remained alive (apoptotic levels similar to *Mock* condition) during the whole experiment (see new Figure 3—figure supplement 4). These findings suggest that α cells are indeed more resistant to CVB infection than β cells. A sentence regarding these results is now included in the subsection headed “Pancreatic α cells are resistant against CVB- but not against cytokine- or double stranded RNA (dsRNA)-induced cell death”.

*6) The main conclusion of the paper is that α cells are only transiently infected with CBVs and can clear the virus infection, while β cells are infected without the ability to eradicate the virus. The stainings performed on pancreata from neonates who died from acute infections may in part support such a conclusion. The authors' conclusions would however be substantially supported if infections of whole pancreatic islets (which the authors should have at hand) would demonstrate that β cells die and α cells survive over time. Also, did the authors count the α cells before and after infection (e.g. 36h)? No change in cell numbers would support the conclusions drawn by the authors. Please provide the data requested*.

These relevant points are well taken. We have now performed new experiments in dispersed human islets infected with CVB5 (M.O.I 10, 8 h of infection; n = 2), where we detected by immunofluorescence the presence of the enteroviral capsid protein VP1 in both α cells (glucagon-positive) and β cells (insulin-positive) (Figure 11). Thus, we found that 52% (Experiment 1) or 33% (Experiment 2) of insulin-positive cells were also positive for VP1, while 28% (Experiment 1) or 27% (Experiment 2) of cells were double positive for glucagon and VP1 (Figure 11). Similar results were observed when CVB4 was used (data not shown). These findings confirm the human islet data described in the Figure 12 of the manuscript. A sentence regarding these results is now included in the subsection “Pancreatic α cells express a vigorous cell-autonomous immune response against viral infection”.

Unfortunately, and due to technical issues, we have not been successful in measuring apoptosis in dispersed human islets infected with CVB5 (M.O.I 10, 24 hr of infection). In these experiments we immunostained dispersed islets for glucagon, insulin and cleaved caspase 3 (used as a marker of apoptosis), but could not discriminate between cleaved caspase 3 staining and background autofluorescence signals under the different experimental conditions tested.

Following the additional suggestion by the referees, we have now carried out new experiments to count the number of α cells before and after viral infection. The number of α cells was counted both in the supernatant and attached to the coverslip after trypsinization. As observed in the new Figure 3—figure supplement 6, the number of α cells in both conditions did not change before or after viral infection, indicating that α cells survived the infection. We have now mentioned these new findings in Results (subsection “Pancreatic α cells are resistant against CVB- but not against cytokine- or double stranded RNA (dsRNA)-induced cell death”).

*7) On a similar note, α and β cells are, according to the Material and methods section, grown in different concentrations of FBS and glucose (α cells: 6.1 mM glucose and 10% FBS; β cells: β cells: 10 mM glucose and 5% FBS). Could the different culture conditions contribute to the differences in susceptibility to infection? Please comment on this, and if possible address*.

To address this relevant question, we have now performed new experiments where FACS-purified α cells were cultured under the same conditions as β cells, i.e. 10 mM glucose and 5% FBS. The results obtained indicate that, regardless of the culture conditions, α cells remain resistant to virus-induced apoptosis (new Figure 3—figure supplement 7). These new findings are now mentioned in Results (please see the subsection “Pancreatic α cells are resistant against CVB- but not against cytokine- or double stranded RNA (dsRNA)-induced cell death”).

*8) The authors use CBV4 and CBV5 to infect cells. These viruses were grown in cell lines. An appropriate control in these experiments would be UV-inactivated virus, as cellular debris including cytokines etc. could have an effect on cell survival. Was such a control included? Please provide data, and if not, undertake studies to show that UV inactivated virus does not impact on cell survival*.

Following the referees’ suggestion, CVB5 stock was inactivated by UV and tested for toxicity on α and β cells. These results are now presented in Figure 3—figure supplement 5, outlined in the Results, and the procedures used indicated in Material and methods, in the subsection headed “Viral infection”.

*9) By PCR it is demonstrated that both α and β cells express receptors for CBVs. More appropriate would have been to analyse surface expression by FACS, as mRNA expression. Please address this or explain why this is not possible*.

In an attempt to detect by FACS the presence of CAR at the cell surface of rat islet cells, we tested two antibodies: a polyclonal rabbit serum (sc-15405; Santa Cruz Biotechnology) and the monoclonal mouse RmcB clone (Merck-Millipore). Unfortunately, none of these antibodies allowed a specific detection of CAR at the surface of rat β cells in both living and paraformaldehyde fixed cells (data not shown). The available antibodies for FACS detection of CAR were raised against sequences of the human protein, which are poorly conserved in the rat protein. This may explain why we did not manage to detect CAR at the surface of rat cells.

As an alternative to the detection of the receptor, we measured virus bound to the cells shortly after infection, which may be more biologically relevant in light of the viral cycle. For this purpose we measured the input virus load in α and β cells two hours after initial exposure. To achieve this, cells were washed 3 times in medium before lysis and titration on GMK cells, done by limit-dilution assay. As presented in Figure 13 the viral titers obtained were similar for α and β cells.

Author response image 1.Viral titers of CVB5 bound to FACS sorted primary rat α or **β** cells. FACS-purified rat α cells (˃90% purity) were infected with CVB5 (M.O.I. 5). After 2 hr incubation cells are washed with medium 3 times before freezing in complete medium. The cell-bound virus was then titrated by limit dilution assay and TCID50 titers calculated according to the Kärber formula. To determine the background of unbound virus, wells devoid of cells were treated the same way and corresponding titers. The unbound viruses were subtracted in each experiment. Results are Mean and SEM of 2-3 independent experiments.**DOI:**
http://dx.doi.org/10.7554/eLife.06990.032

In addition, we detected the rat CAR protein by western blot with the polyclonal sera, and observed equivalent expression of CAR protein in both α and β cells. These results reinforce the idea that the CAR protein is well expressed in both cell types and the data are now presented as Figure 4—figure supplement 1. A sentence regarding these results is now included in the subsection headed “Pancreatic α cells are resistant against CVB- but not against cytokine- or double stranded RNA (dsRNA)-induced cell death”.

In line with the observations described above, the percentages of adeno-GFP positive α and β cells in Figure 5 confirm the presence of functional CAR receptor (used also by the adenovirus) at the surface of both cell types.

Minor comments:

*10) Provide a rationale for why you selected the CBV5 serotype for these experiments and not e.g. CBV1*.

We choose to analyze the effect of CVB5 and CVB4, both serotypes detected in T1D donors (Yeung et al., BMJ 342: d35, 2012), in order to allow comparisons with our previous studies based on infection of β cells with these viral serotypes (Ylipaasto et al., Diabetologia 47: 225-239, 2004; Ylipaasto et al., Diabetologia 48: 1510-1522, 2005; Ylipaasto et al, Diabetologia 55:3273–3283, 2012). Importantly, both CVB4 and CVB5 have been detected in islets of T1D donors (Yeung et al., BMJ 342: d35, 2012) and CVB4 has been associated to diabetes onset (Dotta et al., Proc Natl Acad Sci U S A 104: 5115-5120, 2007; Gallagher et al., Diabetes 64:1358-69, 2015). We did not test CVB1 serotypes in rat primary cells because a study by Nair and collaborators indicate that CVB1 does not multiply in rat β cells (Nair et al., J Med Virol 82:1950–1957, 2010). This information is now provided in Methods, in the subsection headed “Cell treatments and nitric oxide measurement”.

*11) Virus concentrations are given as MOI, but the titers of replicating viruses in cells and media are given as TCID50 making these concentrations difficult to compare. Please, clarify*.

The viral stocks are produced in GMK cells and titrated by plaque assay in GMK cells to provide a titer in pfu/ml. As this titration protocol demands a larger volume of virus, we preferred the titration protocol by limit dilution assay that provides a titer in TCID50. To compare both viral concentrations, we titrated our viral stocks in both ways and obtained similar values as presented in the table below. This observation is now mentioned in Material and methods, in the subsection headed “Viral infection”.Viral stocksTitration in pfu/mlTitration in TCID50/mlCVB5 29/04/20134·10^7^7.9·10^7^4·10^7^3.2·10^7^2.5·10^7^CVB5 10/10/201410^8^3.1·10^8^CVB4 21/05/20143·10^7^6.3·10^7^CVB4 27/09/20141.9·10^7^1.6·10^7^